# SmoothSpike: Spiking Transformer with Learnable Hadamard Transformation

**Zijian Zhou** [1]   **Wenjie Wei** [1]   **Yu Liang** [1]   **Jialin Li** [1]   **Ammar Belatreche** [2]   **Honglin Cao** [1]   **Shuai Wang** [1]
**Malu Zhang** [1 3]   **Yang Yang** [1]   **Haizhou Li** [3 4]

## Abstract

Spiking Neural Networks (SNNs) have attracted growing attention due to their sparse spike-based communication and inherent temporal dynamics. However, their discrete information representation fundamentally limits expressiveness, resulting in a notable performance gap relative to Artificial Neural Networks (ANNs) on language modeling tasks. In this paper, we reveal that this gap is fundamentally rooted in a spike saturation-induced information homogenization problem: within a bounded time window, distinct high-amplitude inputs converge to identical spike counts, compressing neural representations and impairing fine-grained semantic discrimination across layers. To address this, we propose SmoothSpike, which applies a randomized Hadamard transformation to smooth pre-activation inputs and theoretically proves that it bounds the maximum input to $\mathcal{O}(\sqrt{\frac{\log n}{n}})$ with high probability. To further improve adaptability across varying input distributions, we extend the fixed transformation within SmoothSpike to a learnable orthogonal matrix updated via Newton-Schulz iterations, which can be fused into model weights at inference with no additional overhead. Experiments on the GLUE benchmark show that SmoothSpike effectively reduces information homogenization, yielding an 8.2% average improvement over the Spikingformer baseline without compromising the efficiency inherent to spike-driven computation. These results advance the prospects for energy-efficient and high-performance language modeling on edge devices. Code is available at https://github.com/CayleyZ/SmoothSpike.

[1]University of Electronic Science and Technology of China [2]Northumbria University [3]Shenzhen Loop Area Institute [4]The Chinese University of Hong Kong, Shenzhen (CUHK-Shenzhen). Correspondence to: Wenjie Wei <wjwei@std.uestc.edu.cn>.

*Proceedings of the 43$^{rd}$ International Conference on Machine Learning*, Seoul, South Korea. PMLR 306, 2026. Copyright 2026 by the author(s).

## 1. Introduction

Spiking neural networks (SNNs) have gained increasing attention in recent years due to the spike-based communication and temporal dynamics (Gerstner & Kistler, 2002; Maass, 1997). Specifically, SNNs utilize binary spikes to carry information and transmit information in a spike-driven manner, offering significant efficiency advantages. Moreover, the inherent temporal nature of SNNs allows for fine-grained representation of sequential data, making them well-suited for spatiotemporal modeling. As a result, a growing body of research is exploring the use of SNNs for complex language modeling tasks, such as SpikingBERT (Bal & Sengupta, 2024), SpikeLM (Xing et al., 2024b), and SpikeBERT (Lv et al., 2025). However, due to the discrete spikes, SNNs are less expressive in feature representation compared to Artificial Neural Networks (ANNs), resulting in a noticeable performance gap in complex language modeling tasks.

To narrow the accuracy gap, several studies have focused on improving the representational capacity of SNNs (Guo et al., 2022; Fang et al., 2023; Xiao et al., 2025; Wang et al., 2025c). For example, Guo et al. (2024); Xing et al. (2024b); Wang et al. (2025a) extend traditional binary spikes to ternary spikes, introducing negative values to enhance the representation richness. In addition, Luo et al. (2024); Qiu et al. (2025) introduce multi-level spike states in the spike generation mechanism to provide more fine-grained discrete amplitudes. Despite the improved performance, these methods primarily enhance SNNs' representational capacity by expanding the set of possible spike values. Nevertheless, these studies overlook the fact that within a limited time window, a single spiking neuron has a clear upper bound on the values it can represent, which directly limits its information discriminability and model representation capacity.

In SNNs, each spiking neuron has an upper limit $T$ on the number of spikes it can emit due to the constrained time window. When the input value is large, the output spike count may reach this limit. As a result, different high-amplitude inputs may produce nearly identical output responses, making it difficult to distinguish input differences in the output space (Sava et al., 2023). We refer to this phenomenon as the spike saturation-induced information homogenization problem. This is equivalent to truncating the dynamic range

of the output, thereby compressing the representation of spiking neurons. This limitation hinders the ability of SNNs to capture fine-grained semantic differences. Therefore, addressing the spike saturation issue is crucial for improving SNN performance in complex language modeling tasks.

In this paper, we propose the SmoothSpike, aimed at alleviating the spike saturation-induced information homogenization to improve performance. Specifically, we first analyze the spike saturation issue in spiking transformers and experimentally reveal the resulting information homogenization phenomenon. To address this, we introduce a randomized Hadamard matrix to smooth the inputs of spiking neurons, reducing both the likelihood of neuron saturation and the dynamic range of inputs to saturated neurons. We then replace the randomized Hadamard matrix with a learnable orthogonal transformation matrix, which better adapts to varying input ranges, further enhancing the representational capacity of SNNs. We summarize our contributions as follows:

- We analyze the spike saturation issue, highlighting its widespread occurrence across layers and revealing the induced information homogenization phenomenon. This prevents the differences in input distribution from being reflected in the output space, weakening SNNs' representational capacity and degrading performance.

- We propose using a randomized Hadamard transformation to smooth the inputs of spiking neurons and provide a theoretical proof that this transformation can constrain the maximum element of the input vector to a range of $\mathcal{O}(\sqrt{\frac{\log n}{n}})$. By applying this transformation, both the likelihood of spike saturation and the variance of inputs to saturated neurons are effectively reduced, thus preserving the model's representational capacity.

- We propose a learnable orthogonal transformation method, which is initialized with a randomized Hadamard matrix and approximates the orthogonal constraint using the Newton-Schulz iteration. This method allows the transformation matrix to adapt to varying input distributions during training, further enhancing the representational capacity of SNNs.

- Experiments on the GLUE benchmark show that SmoothSpike effectively alleviates the spike saturation issue, leading to substantial performance improvements. Furthermore, detailed ablation studies confirm the effectiveness and generalizability of our method.

## 2. Related work

### 2.1. SNNs for language modeling

Given the energy efficiency and temporal dynamics of SNNs (Hu et al., 2015; Tan et al., 2013; Tang et al., 2024), an in-

creasing number of studies are exploring their application in language modeling tasks (Zhang et al., 2025b; 2024; 2025a). For example, SpikingBERT (Bal & Sengupta, 2024) proposes a spiking language model that utilizes spiking attention and knowledge distillation, achieving energy-efficient training and improved performance. SNN-BERT (Su et al., 2024) introduces a bidirectional parallel spiking neuron architecture and individual-based coding to build a parallel SNN, reducing memory overhead. Sorbet (Tang et al., 2025) proposes power-of-two softmax and bit-shifting powerNorm to replace traditional energy-intensive softmax and layer normalization, resulting in significant energy savings. SpikeLM (Xing et al., 2024b) introduces an elastic bi-spiking mechanism for language modeling, using bidirectional spike encoding, spike frequency, and amplitude encoding to enhance performance. SpikeBERT (Lv et al., 2025) improves the Spikformer for language tasks by introducing a two-stage knowledge distillation method, enabling energy-efficient text classification. While these works advance SNN-based language modeling, most do not explicitly analyze or address the limited representational power of spiking neurons, which we identify as the main bottleneck for performance. Notably, SpikeLM partially mitigates this limitation, but at the cost of compromising the spike-driven computation that underpins SNN efficiency.

### 2.2. Spiking neurons with high expressive capacity

Several studies have focused on improving the representational capacity of SNNs to enhance their task performance (Fang et al., 2023; Feng et al., 2022; Wei et al., 2024a; 2025). For example, (Guo et al., 2024) proposes the ternary spike neuron, which extends binary spike activation to values of -1, 0, and 1, thereby increasing the information capacity of SNNs while maintaining their energy efficiency. (Wang et al., 2025a) introduces a ternary spike-based neuromorphic signal processing system that uses a threshold-adaptive encoding method and a quantized ternary SNN, achieving energy-efficient signal processing and improved performance in tasks like keyword recognition and EEG identification. In addition to introducing negative values to enhance representational richness, some approaches use multi-level spike states to provide finer discrete amplitudes. For example, (Luo et al., 2024) presents the integer LIF neuron model, which reduces quantization errors during training with integer-valued activations and preserves spike-driven inference during testing, significantly improving object detection performance in SNNs. (Qiu et al., 2025) integrates an information-enhanced LIF neuron into a spiking transformer, achieving state-of-the-art results in various vision tasks. Despite the improvement, these studies overlook the fact that a single spiking neuron has a clear upper bound on the values it can represent, which directly limits its information discriminability and model representation capacity.

## 3. Preliminary

### 3.1. Spiking Neurons

Spiking neurons are the basic computational units of SNNs (Zhang et al., 2021; Wei et al., 2024b). Unlike conventional artificial neurons that output continuous-valued activations, spiking neurons communicate through binary events over discrete time steps. Given a time window of length $T$, a spiking neuron maintains an internal membrane potential $V_t$ and receives an input current $x_t$ at each time step $t$. Its dynamics can be written as:

$$V_t = \Phi(V_{t-1}, x_t), \qquad s_t = \mathcal{H}(V_t - \vartheta), \qquad (1)$$

where $\Phi(\cdot)$ denotes the membrane potential update function, $\vartheta$ is the firing threshold, $\mathcal{H}(\cdot)$ is the Heaviside step function, and $s_t \in \{0, 1\}$ indicates whether a spike is emitted at time step $t$. After spike firing, the membrane potential is typically reset via a model-dependent mechanism. For brevity, we denote the spiking neuron operation as $\mathcal{SN}(\cdot)$.

### 3.2. Hadamard matrix

A Hadamard matrix $\mathbf{H}_n$ is a square matrix whose entries are in $\{-1, +1\}$ and whose rows are mutually orthogonal. Denoting the $i$-th row of $\mathbf{H}_n$ as $\mathbf{h}_i$, we have:

$$\mathbf{h}_i^\top \mathbf{h}_j = n \cdot \mathbf{1}[i = j], \quad \forall i, j \in [n], \qquad (2)$$

where $\mathbf{1}[\cdot]$ is the indicator function that equals 1 if the condition holds and 0 otherwise. Equivalently, this orthogonality can be written in matrix form as $\mathbf{H}_n^\top \mathbf{H}_n = n\mathbf{I}_n$. By applying the normalization factor $\frac{1}{\sqrt{n}}$ for $\mathbf{H}_n$, we obtain $\bar{\mathbf{H}} = \frac{1}{\sqrt{n}}\mathbf{H}_n$, which is an orthogonal matrix satisfying:

$$\bar{\mathbf{H}}^\top \bar{\mathbf{H}} = \mathbf{I}_n, \quad \bar{\mathbf{H}}^{-1} = \bar{\mathbf{H}}^\top. \qquad (3)$$

$\bar{\mathbf{H}}$ defines an orthogonal transformation that preserves the Euclidean norm of any input $\mathbf{x} \in \mathbb{R}^n$, i.e., $\|\bar{\mathbf{H}}\mathbf{x}\|_2 = \|\mathbf{x}\|_2$.

The structure of $\bar{\mathbf{H}}$ provides two practically important properties. First, $\bar{\mathbf{H}}$ spreads the energy of any input vector uniformly across all dimensions while preserving its $\ell_2$ norm. This ensures sparsely distributed outlier values in the input are suppressed after transformation, yielding a more uniform entry distribution. Second, since $\bar{\mathbf{H}} = \frac{1}{\sqrt{n}}\mathbf{H}_n$ with $\mathbf{H}_n \in \{\pm 1\}^{n \times n}$, the main computation of $\bar{\mathbf{H}}\mathbf{x}$ reduces to additions and subtractions in $\mathbf{H}_n\mathbf{x}$, with the factor $\frac{1}{\sqrt{n}}$ applied once or absorbed into subsequent parameters. These properties make Hadamard transformation well-suited for quantization and model compression in deep learning (Xiao et al., 2023; Liu et al., 2025b). For example, multiplying weights or activations by $\bar{\mathbf{H}}$ can significantly reduce quantization error and suppress outliers at relatively low computational cost (Tseng et al., 2024; Ashkboos et al., 2024).

Motivated by this, we leverage the Hadamard transformation to make the inputs of spiking neurons more evenly distributed, thereby mitigating the spike saturation problem.

### 3.3. Root mean square layer normalization

Layer normalization is a standard normalization technique in Transformer. Existing spike-based language models typically follow the original design by adopting LayerNorm with a post-norm architecture (Xing et al., 2024b; Tang et al., 2025; Wang et al., 2025b). In this work, we instead employ root mean square layer normalization (RMSNorm) (Zhang & Sennrich, 2019) with a pre-norm architecture (Xiong et al., 2020), which is central to our Hadamard-based transformation framework. Given an input vector $\mathbf{x} \in \mathbb{R}^n$, RMSNorm normalizes $\mathbf{x}$ by its root-mean-square value:

$$\text{RN}(\mathbf{x}) = \frac{\mathbf{x}}{\sqrt{\frac{1}{n}\sum_{i=1}^n x_i^2}} \odot \boldsymbol{\gamma}, \qquad (4)$$

where $\boldsymbol{\gamma} \in \mathbb{R}^n$ is a learnable channel-wise scaling parameter, and $\odot$ denotes element-wise multiplication. In the pre-norm architecture, RMSNorm is followed by spiking neurons, allowing $\boldsymbol{\gamma}$ to be absorbed into their firing thresholds. The details of this fusion are provided in Appendix C.

Once the learnable scaling parameter $\boldsymbol{\gamma}$ is absorbed into the firing thresholds, RMSNorm reduces to a scale-free $\ell_2$ rescaling operation, which we denote by $\text{RN}_0(\cdot)$. This endows the RMSNorm with orthogonal equivariance. More specifically, for any orthogonal matrix $\mathbf{Q}$ and weight matrix $\mathbf{W}$, denoting $\mathbf{z} = \mathbf{W}\mathbf{x}$, we can obtain:

$$\mathbf{Q}\,\text{RN}_0(\mathbf{z}) = \frac{\mathbf{Q}\mathbf{z}}{\sqrt{\frac{1}{n}\|\mathbf{z}\|_2^2}} \overset{(*)}{=} \frac{\mathbf{Q}\mathbf{z}}{\sqrt{\frac{1}{n}\|\mathbf{Q}\mathbf{z}\|_2^2}} = \text{RN}_0(\mathbf{Q}\mathbf{z}),$$

$$(5)$$

where $(*)$ holds because $\|\mathbf{Q}\mathbf{z}\|_2 = \|\mathbf{z}\|_2$ for any orthogonal $\mathbf{Q}$. This equivariance property provides the theoretical basis for absorbing the Hadamard matrix directly into the weight matrices in our SmoothSpike, thereby avoiding additional computational overhead during inference.

## 4. Methodology

In this section, we first analyze the spike saturation and information homogenization issue. Then, we use a randomized Hadamard transform to address this issue with theoretical support. Finally, we extend it to a learnable orthogonal matrix for adaptive smoothing across input distributions.

### 4.1. Problem Analysis

Recent years have witnessed growing interest in exploring SNNs for language modeling tasks. Despite notable progress, these studies exhibit a clear performance gap relative to ANNs. Below, we analyze this bottleneck in detail.

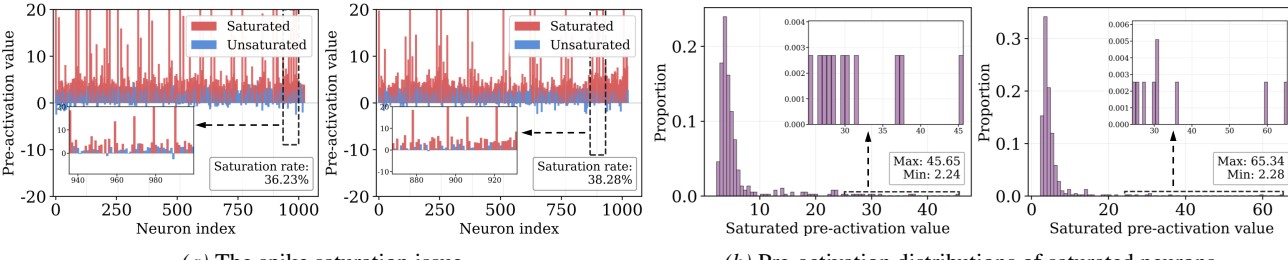

*(a)* The spike saturation issue.   *(b)* Pre-activation distributions of saturated neurons.

*Figure 1.* Visualization of the spike saturation problem and its induced information homogenization. (a) A large proportion of spiking neurons are saturated. (b) Pre-activation distributions of saturated neurons, which span a wide range but produce identical spike trains.

The performance bottleneck of spike-based language models stems from an intrinsic constraint of spiking neurons: within a time window of length $T$, each neuron can emit at most $T$ spikes. Formally, let $V_t$ denote the membrane potential of a spiking neuron at time step $t$. For two distinct input currents $x_1 > x_2 \gg 0$, both may drive the neuron to its maximum firing rate, yielding identical spike counts:

$$\sum_{t=1}^{T} \mathcal{SN}(x_1, V_{t-1}^{(1)}) = \sum_{t=1}^{T} \mathcal{SN}(x_2, V_{t-1}^{(2)}) = T, \quad (6)$$

In this case, neurons receive distinct inputs but produce identical output spikes. We refer to this as the spike saturation problem. To illustrate its extent, we use Spikingformer-12-768 evaluated on the CoLA task as a case study and visualize saturated neurons across two layers. As shown in Figure 1a, a large proportion of neurons reach their maximum firing rate, showing that spike saturation is a pervasive issue.

The spike saturation problem forces spiking neurons to produce identical spike trains regardless of their inputs, thereby rendering semantically distinct signals indistinguishable. We refer to this phenomenon as spike saturation-induced information homogenization. To prove this, we further analyze the input distribution of the saturated neurons identified in Figure 1a. As shown in Figure 1b, the pre-activation inputs to saturated neurons span a wide dynamic range, from 2.24 to 65.34, yet are all mapped to identical output spike trains. In short, spiking neurons collapse diverse high-magnitude inputs into indistinguishable representations, severely impairing the network's ability to capture fine-grained semantic distinctions. More critically, this loss of discriminability may further accumulate with network depth, progressively amplifying performance degradation on language modeling tasks. Therefore, reducing the occurrence of spike saturation is crucial for preserving SNNs' representational capacity and improving performance.

### 4.2. Randomized Hadamard transform for mitigating spike saturation

As analyzed above, the spike saturation issue arises from high-magnitude inputs that drive neurons to their maximum

firing rate, collapsing distinct inputs into identical output spike trains. Fortunately, the Hadamard matrix, described in Sec. 3.2, can uniformly redistribute the energy of any input vector across all dimensions while preserving its $\ell_2$ norm, thereby naturally suppressing outlier values that may drive neurons into saturation. This motivates us to apply a randomized Hadamard transform to the pre-activation inputs of spiking neurons to reduce the probability of saturation.

Let $\mathbf{x}^\ell \in \mathbb{R}^n$ the pre-activation input of spiking neurons at layer $\ell$, we define the randomized Hadamard matrix as

$$\mathbf{H}_{\mathrm{R}} = \bar{\mathbf{H}}\mathbf{S}, \quad (7)$$

where $\mathbf{S} = \mathrm{diag}(\sigma_1, \ldots, \sigma_n)$ is a random diagonal matrix with independent Rademacher entries, i.e., $\mathbb{P}(\sigma_i = 1) = \mathbb{P}(\sigma_i = -1) = 1/2$. The transformed input is then given by

$$\tilde{\mathbf{x}}^\ell = \mathbf{H}_{\mathrm{R}}\mathbf{x}^\ell. \quad (8)$$

From this formulation, the maximum element of $\tilde{\mathbf{x}}^\ell$ is suppressed through two mechanisms. First, each input component is scaled by a factor of small magnitude, i.e., $1/\sqrt{n}$; for a typical hidden dimension such as $n = 768$, this factor is about $0.04$. Second, the scaling coefficients take both positive and negative values, allowing input components to partially cancel upon summation partially. Together, these two mechanisms prevent large values from concentrating in a few dimensions and disperse them more uniformly across all dimensions. Thus, extreme pre-activation values that would saturate individual neurons are smoothed out, reducing spike saturation and alleviating information homogenization. This preserves the model's capacity for fine-grained semantic discrimination, improving performance.

To recover representations compatible with subsequent network layers, we apply the inverse Hadamard transform to neuron outputs. Since $\mathbf{H}_{\mathrm{R}}$ is orthogonal, its inverse equals its transpose, i.e., $\mathbf{H}_{\mathrm{R}}^{-1} = \mathbf{H}_{\mathrm{R}}^{\top}$. Letting $\mathbf{s}^\ell = \mathcal{SN}(\tilde{\mathbf{x}}^\ell)$ denote the output spike train, the final output of layer $\ell$ is:

$$\mathbf{y}^\ell = \mathbf{H}_{\mathrm{R}}^{\top}\mathbf{s}^\ell. \quad (9)$$

Together, Eq. 8 and Eq. 9 form an encoding-decoding pair: the encoding stage projects inputs into a space better suited

for spiking neuron processing, while the decoding stage maps the outputs back to the original representation space, enabling seamless integration with the remainder of the network. This ensures that the overall network information flow is not disrupted when the Hadamard transform is applied.

We further provide a theoretical analysis to explain why the randomized Hadamard transform can mitigate spike saturation. Specifically, we derive an upper bound on the maximum element of the transformed vector $\tilde{\mathbf{x}}^\ell$, showing that this transform suppresses extreme pre-activation values with high probability. This is formally stated as follows.

**Theorem 4.1.** *(Maximum Element Bound for Randomized Hadamard Transform) Let* $\mathbf{H}_R \in \mathbb{R}^{n \times n}$ *be a randomized Hadamard matrix and* $\mathbf{x}^\ell \in \mathbb{R}^n$ *the pre-activation input at layer* $\ell$. *The transformed input is given by* $\tilde{\mathbf{x}}^\ell = \mathbf{H}_R \mathbf{x}^\ell$. *Let* $\mathbf{e}_i \in \mathbb{R}^n$ *denote the* $i$-*th standard basis vector. Then, for any* $\epsilon \in (0, 1)$, *the following bound holds:*

$$P\left( \max_i \left| \mathbf{e}_i^T \tilde{\mathbf{x}}^\ell \right| < \|\mathbf{x}^\ell\|_2 \sqrt{\frac{2}{n} \log \frac{2n}{\epsilon}} \right) \geq 1 - \epsilon. \quad (10)$$

Theorem 4.1[1] shows that, after the randomized Hadamard transform, the maximum value of $\tilde{\mathbf{x}}^\ell$ is bounded by $\mathcal{O}\left( \sqrt{\frac{\log n}{n}} \right)$ with high probability. This provides a rigorous justification for the effect of Eq. 8. We use an example to understand the practical implications of this theorem. Consider an extreme case where one value of $\mathbf{x}^\ell$ dominates and approaches $\|\mathbf{x}^\ell\|_2$. Without the transformation, this large value may directly drive the corresponding spiking neuron into saturation. In contrast, after applying the randomized Hadamard transform $\mathbf{H}_R$, the maximum element of $\tilde{\mathbf{x}}^\ell$ is bounded by $\|\mathbf{x}^\ell\|_2 \sqrt{\frac{2 \log(2n/\epsilon)}{n}}$ with probability at least $1 - \epsilon$. For a common setting in SNNs, i.e., $n = 768$ and $\epsilon = 0.01$, this bound is approximately $0.18 \cdot \|\mathbf{x}^\ell\|_2$. This is far below the pre-transformation extremes, indicating that the Hadamard transform can effectively suppress extreme inputs and reduce spike saturation.

### 4.3. Learnable Orthogonal Transform Matrix

While a fixed randomized Hadamard matrix effectively reduces spike saturation, it may not be optimal for inputs with different distributions. As shown in Figure 2, the input features before the Hadamard transforms show clear layer-wise distribution differences. This suggests that the transform matrix should be adapted to each layer's distribution. Therefore, we initialize the transform matrix with a randomized Hadamard matrix and optimize it during training to better align with the data distribution, potentially improving model performance. Crucially, the transform matrix must remain

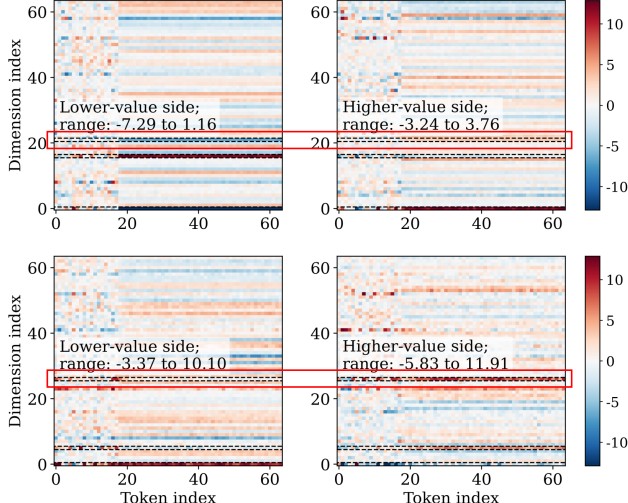

*Figure 2.* Comparison of input features before the Hadamard transform across layers. Each plot shows a subset of inputs in a specific layer, and black dashed boxes highlight the largest differences between the two layers in each row.

orthogonal throughout training, since orthogonality guarantees the equivariance of RMSNorm to the transform, thereby enabling the transform to be seamlessly fused into model weights during inference.

Several methods exist for enforcing orthogonality in learnable matrices. We adopt the matrix sign function (msign) for orthogonal projection, motivated by a key theoretical guarantee: it yields the optimal orthogonal approximation of any given matrix. This is formally stated as follows.

**Theorem 4.2.** *(Optimal Orthogonal Approximation Property of Matrix Sign Function) For any full-rank matrix* $\widetilde{\mathbf{H}} \in \mathbb{R}^{n \times n}$, *the matrix sign function* $\mathbf{H}$ *is the optimal orthogonal approximation of* $\widetilde{\mathbf{H}}$ *under the Frobenius norm:*

$$\mathbf{H} = \mathrm{msign}(\widetilde{\mathbf{H}}) = \arg\min_{\mathbf{H}' \in \mathcal{O}(n)} \|\widetilde{\mathbf{H}} - \mathbf{H}'\|_F, \quad (11)$$

*where* $\mathcal{O}(n)$ *is the* $n$-*dimensional orthogonal matrix group.*

Theorem 4.2[2] ensures that, during training, gradient updates can freely optimize the learnable matrix $\widetilde{\mathbf{H}}$, while msign projects it onto the nearest orthogonal matrix in each forward pass. This decoupling preserves optimization flexibility while strictly enforcing orthogonality.

The msign operation is defined via singular value decomposition (SVD). For a matrix $\widetilde{\mathbf{H}} \in \mathbb{R}^{n \times n}$ with singular value decomposition $\widetilde{\mathbf{H}} = \mathbf{U}\mathbf{\Sigma}\mathbf{V}^\top$, it is formally defined as:

$$\mathrm{msign}(\widetilde{\mathbf{H}}) = \mathbf{U}\mathbf{V}^\top, \quad (12)$$

which replaces all singular values with 1 while preserving the singular vectors. Despite the optimality of the msign

---

[1]The proof of Theorem 4.1 is provided in Appendix A.1.

[2]The proof of Theorem 4.2 is provided in Appendix A.2.

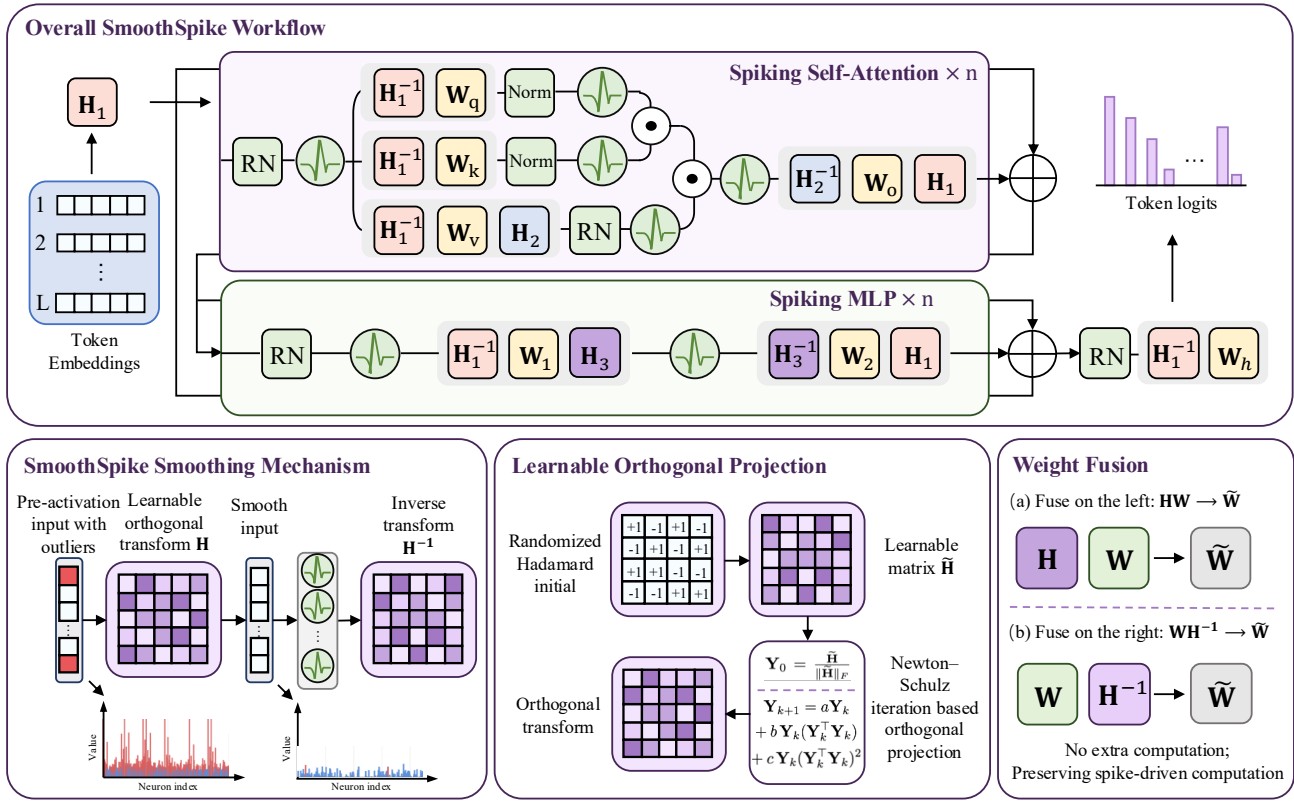

*Figure 3.* Overall architecture of the proposed SmoothSpike, illustrated using the Spikingformer (Zhou et al., 2026) framework.

function, directly computing it via SVD is computationally expensive, as SVD scales cubically with matrix dimension. Furthermore, SVD-based computation may introduce extra challenges for stable and efficient gradient backpropagation in modern deep learning frameworks.

We then approximate the msign function using a fifth-order Newton–Schulz iteration in practice (Liu et al., 2025a). The iteration is initialized as $\mathbf{Y}_0 = \frac{\widetilde{\mathbf{H}}}{\|\widetilde{\mathbf{H}}\|_F}$, where $\|\cdot\|_F$ is the Frobenius norm, which ensures the singular values of $\mathbf{Y}_0$ lie in $(0, \sqrt{3})$. The iteration proceeds as:

$$\mathbf{Y}_{k+1} = a\mathbf{Y}_k + b\,\mathbf{Y}_k(\mathbf{Y}_k^\top \mathbf{Y}_k) + c\,\mathbf{Y}_k(\mathbf{Y}_k^\top \mathbf{Y}_k)^2, \quad (13)$$

After $K$ iterations, $\mathbf{Y}_K$ converges to an orthogonal approximation of $\widetilde{\mathbf{H}}$. Further details on the Newton–Schulz iteration and gradient backpropagation are provided in Appendix B.

In summary, we use the Newton–Schulz iteration to efficiently approximate the matrix sign function, maintaining the orthogonal projection of the learnable matrix during training. This offers two key advantages. First, it is more efficient than direct SVD, requiring less execution time and fewer computational resources. Second, its computation graph is fully differentiable, allowing seamless integration with modern automatic differentiation frameworks and enabling end-to-end gradient-based optimization.

### 4.4. Overall Architecture of the SmoothSpike

SmoothSpike can be incorporated into any spiking transformer architecture to improve its representational capacity and task performance. Below, we illustrate its implementation using Spikingformer (Zhou et al., 2026) as an example, shown in Fig. 3. First, we adopt a pre-norm architecture and replace batch normalization with RMSNorm. This is essential for fusing the learnable orthogonal transforms into model weights, as scale-free RMSNorm satisfies the orthogonal equivariance property discussed in Sec. 3.3. Second, we insert learnable orthogonal transformation matrices at the pre-activation inputs of spiking neurons to smooth input distributions and alleviate spike saturation. Concretely, we share $\mathbf{H}_1$ across residual-connected branches to keep their representation spaces aligned, apply $\mathbf{H}_2$ to the value projection branch, and use a block-diagonal $\mathbf{H}_3$ with four equally sized blocks in the MLP to reduce training cost. LIF neurons for query and key generation are left untransformed, as they exhibit only mild saturation empirically. The complete training procedure is summarized in Algorithm 1.

In SmoothSpike, all transformation matrices can be absorbed into adjacent weight matrices during inference, thereby adding no extra computational cost while improving performance. Specifically, the matrix $\mathbf{H}_3$ can be directly

absorbed into adjacent linear weights. For $\mathbf{H}_1$ and $\mathbf{H}_2$, the fusion is enabled by the orthogonal equivariance property of RMSNorm. Taking $\mathbf{H}_1$ as an example, its participation in the forward computation can be expressed as

$$\widetilde{\mathbf{x}}^\ell = \mathbf{H}_1\,\mathrm{RN}_0(\mathbf{W}^\ell\mathbf{y}^{\ell-1}). \qquad (14)$$

Since $\mathbf{H}_1$ is orthogonal, $\mathrm{RN}_0(\cdot)$ is equivariant to this transformation. This expression can thus be rewritten as

$$\widetilde{\mathbf{x}}^\ell = \mathrm{RN}_0(\mathbf{H}_1\mathbf{W}^\ell\mathbf{y}^{\ell-1}) = \mathrm{RN}_0(\widetilde{\mathbf{W}}^\ell\mathbf{y}^{\ell-1}), \qquad (15)$$

where $\widetilde{\mathbf{W}}^\ell = \mathbf{H}_1\mathbf{W}^\ell$ is the fused weight matrix. This shows that the input-side orthogonal transform can be pre-fused into the corresponding linear projection. Similarly, the inverse transform $\mathbf{H}^\top$ applied after the spiking neurons can be absorbed into the weights of the subsequent linear layer. Consequently, all transformation modules are eliminated after fusion, leaving the original event-driven computation paradigm intact during inference.

## 5. Experiments

### 5.1. Performance on the GLUE Benchmark

To evaluate the effectiveness of our proposed SmoothSpike method, we conduct extensive experiments on the GLUE benchmark (Wang et al., 2018), which comprises eight diverse natural language understanding tasks: MNLI, QQP, QNLI, SST-2, CoLA, STS-B, MRPC, and RTE. Detailed experimental settings are provided in Appendix D. We integrate SmoothSpike into widely used spiking Transformer baselines, and compare their performances against vanilla baselines, as well as other representative SNNs, ANNs, and quantized neural networks (QNNs). The comparison includes BERT$_{\text{base}}$ (Devlin et al., 2019) as a high-performance ANN reference, ELMo (Peters et al., 2018) as an embedding-based baseline, and various quantized and spiking models such as BiBERT (Qin et al., 2022), BiT (Liu et al., 2022), BiPFT (Xing et al., 2024a), SpikeBERT, PSN-BERT, LIF-BERT (Xing et al., 2024b), Spikingformer (Zhou et al., 2026) and SpikeLM.

As shown in Table 1, the vanilla Spikingformer achieves an average score of 66.8 across the GLUE tasks. Incorporating SmoothSpike yields substantial improvements, increasing the average score to 75.0 and achieving a gain of 8.2 points. Moreover, SmoothSpike brings consistent performance improvements across all GLUE tasks. Notable enhancements are observed on tasks requiring fine-grained semantic discrimination. In particular, the scores on CoLA, STS-B, and QNLI are improved by 16.8, 24.8, and 7.8 points, respectively. These gains underscore SmoothSpike's ability to alleviate spike saturation and enhance representational capacity. Similar improvements are observed for SpikeLM, another baseline model, although the improvement is smaller

**Algorithm 1** Overview of the SmoothSpike training scheme and weight fusion applied during inference.

---
**Require:** Spiking Transformer $f_\Theta$; training set $\mathcal{D}$; transform positions $\mathcal{P} = \{1, 2, 3\}$; Newton–Schulz steps $K$; coefficients $a, b, c$; learning rate $\eta$
**Ensure:** Fused SmoothSpike model for inference
1: Initialize each learnable transform $\widetilde{\mathbf{H}}_p^\ell$ with a randomized Hadamard matrix $\mathbf{H}_{\mathrm{R}}$
2: Use block-diagonal structure for $\widetilde{\mathbf{H}}_3^\ell$
3: **for** each mini-batch $(\mathbf{x}, \mathbf{y}) \in \mathcal{D}$ **do**
4:    **for** each layer $\ell$ and transform position $p \in \mathcal{P}$ **do**
5:       Set $\mathbf{Y}_0 = \widetilde{\mathbf{H}}_p^\ell / \|\widetilde{\mathbf{H}}_p^\ell\|_F$
6:       **for** $k = 0$ to $K - 1$ **do**
7:          $\mathbf{Y}_{k+1} = a\mathbf{Y}_k + b\mathbf{Y}_k(\mathbf{Y}_k^\top\mathbf{Y}_k) + c\mathbf{Y}_k(\mathbf{Y}_k^\top\mathbf{Y}_k)^2$
8:       **end for**
9:       Set $\mathbf{H}_p^\ell = \mathbf{Y}_K$
10:    **end for**
11:    **for** each selected spiking branch **do**
12:       Smooth pre-activation input: $\tilde{\mathbf{x}}_p^\ell = \mathbf{H}_p^\ell\mathbf{x}_p^\ell$
13:       Generate spikes: $\mathbf{s}_p^\ell = \mathcal{SN}(\tilde{\mathbf{x}}_p^\ell)$
14:       Decode outputs: $\mathbf{y}_p^\ell = (\mathbf{H}_p^\ell)^\top\mathbf{s}_p^\ell$
15:    **end for**
16:    Skip the transform for query/key LIF neurons
17:    Update $\Theta$ and $\{\widetilde{\mathbf{H}}_p^\ell\}$ by minimizing $\mathcal{L}(f_\Theta(\mathbf{x}), \mathbf{y})$
18: **end for**
19: **for** each trained transform $\widetilde{\mathbf{H}}_p^\ell$ **do**
20:    Recompute $\mathbf{H}_p^\ell = \mathrm{msign}(\widetilde{\mathbf{H}}_p^\ell)$
21:    Fuse input-side transform into weights: $\widetilde{\mathbf{W}}_p^\ell = \mathbf{H}_p^\ell\mathbf{W}_p^\ell$
22:    Fuse output-side inverse transform into next weights: $\widetilde{\mathbf{W}}_{\mathrm{next}}^\ell = \mathbf{W}_{\mathrm{next}}^\ell(\mathbf{H}_p^\ell)^\top$
23: **end for**
24: Remove explicit transform modules after fusion
25: **return** The fused SmoothSpike model

---

than that for Spikingformer. This difference arises because SpikeLM already utilizes advanced mechanisms such as the elastic bi-spiking mechanism to substantially enhance its representational capacity, thereby limiting the additional gains that SmoothSpike can provide.

### 5.2. Ablation Study

To investigate the individual contributions of the components in SmoothSpike, we conduct ablation experiments on the GLUE development set using a 4-layer Spikingformer model with a hidden dimension of 768. The results are presented in Table 2, where we systematically incorporate the Hadamard transformations $\mathbf{H}_1$, $\mathbf{H}_2$, and $\mathbf{H}_3$, and evaluate the impact of making these matrices learnable.

The baseline without any transformations achieves an average score of 64.0. Incorporating the learnable $\mathbf{H}_1$ improves

*Table 1.* Comparison of performance on the GLUE development set for the baseline Spiking Transformer with and without SmoothSpike, alongside various other SNNs, ANNs and QNNs. T refers to time steps of the model. The unit of energy is mJ.

| Model | T | Energy | MNLI$_{m/mm}$ | QQP | QNLI | SST-2 | CoLA | STS-B | MRPC | RTE | Avg. |
|---|---|---|---|---|---|---|---|---|---|---|---|
| BERT$_{base}$ | – | 51.41 | 83.8/83.4 | 90.5 | 90.7 | 92.3 | 60.0 | 89.4 | 89.8 | 69.3 | 83.2 |
| ELMo | – | – | 68.6/– | 86.2 | 71.1 | 91.5 | 44.1 | 70.4 | 76.6 | 53.4 | 70.2 |
| BiBERT | – | – | 66.1/67.5 | 84.8 | 72.6 | 88.7 | 25.4 | 33.6 | 72.5 | 57.4 | 63.2 |
| BiT | – | – | 77.1/77.5 | 82.9 | 85.7 | 87.7 | 25.1 | 71.1 | 79.7 | 58.8 | 71.0 |
| BiPFT | – | – | 69.5/70.6 | 83.7 | 81.7 | 86.2 | 22.9 | 80.2 | 76.2 | 66.1 | 70.8 |
| SpikeBERT | 4 | 14.30 | 71.4/71.0 | 68.2 | 66.4 | 85.4 | 16.9 | 18.7 | 82.0 | 57.5 | 59.7 |
| PSN-BERT | 4 | – | 35.4/35.2 | 0.0 | 50.5 | 50.9 | 0.0 | 6.8 | 81.2 | 52.7 | 34.7 |
| LIF-BERT | 4 | 7.98 | 56.8/55.2 | 70.0 | 60.6 | 80.6 | 14.6 | 20.0 | 82.3 | 53.8 | 54.9 |
| Spikingformer | 4 | 6.76 | 71.9/72.5 | 84.7 | 76.0 | 87.2 | 24.4 | 54.5 | 79.7 | 55.6 | 66.8 |
| **+SmoothSpike** | 4 | 9.45 | **75.1/76.1** | **87.6** | **83.8** | **88.6** | **41.2** | **79.3** | **84.8** | **58.5** | **75.0** |
| SpikeLM | 1 | 3.98 | 76.0/76.9 | 84.0 | 84.9 | 86.5 | 37.9 | **84.3** | 85.6 | **65.3** | 75.7 |
| **+SmoothSpike** | 1 | 7.03 | **76.8/77.7** | **84.3** | **86.8** | **89.5** | **52.7** | 83.5 | **88.1** | 58.1 | **77.5** |

*Table 2.* Ablation results for individual components of SmoothSpike on the GLUE development set. *Learnable* indicates whether the Hadamard matrix is set as learnable.

| Method | | | | MNLI$_{m/mm}$ | QQP | QNLI | SST-2 | CoLA | STS-B | MRPC | RTE | Avg. |
|---|---|---|---|---|---|---|---|---|---|---|---|---|
| H$_1$ | H$_2$ | H$_3$ | Learnable | | | | | | | | | |
| | | | | 67.13/67.37 | 76.82 | 70.73 | 83.72 | 19.10 | 51.84 | 82.27 | 57.04 | 64.00 |
| ✓ | | | | 67.83/68.45 | 78.19 | 71.13 | 83.03 | 19.24 | 50.55 | 81.48 | 53.79 | 63.74 |
| ✓ | | | ✓ | 70.21/69.69 | 79.45 | 75.51 | 84.75 | 16.07 | 61.29 | 81.79 | 54.15 | 65.88 |
| ✓ | ✓ | | ✓ | 71.16/71.30 | 80.24 | 77.06 | 86.01 | 22.30 | 64.18 | 82.04 | 54.51 | 67.64 |
| ✓ | ✓ | ✓ | ✓ | 71.20/71.40 | 80.92 | 78.58 | 85.55 | 19.89 | 65.26 | 82.15 | 55.96 | **67.88** |

the average score to 65.88, yielding a gain of 1.88 points and demonstrating its effectiveness in mitigating saturation. The successive addition of learnable H$_2$ and H$_3$ further boosts the average score to 67.64 and 67.88, corresponding to improvements of 3.64 and 3.88 points over the baseline, respectively. These results indicate the complementary benefits of applying transformations at multiple positions.

To validate the efficacy of our proposed learnable orthogonal matrix strategy, we compare two configurations for H$_1$: (1) a fixed randomized Hadamard matrix, and (2) a learnable matrix initialized with a randomized Hadamard matrix. The learnable H$_1$ outperforms its fixed counterpart by 2.14 points in average score, underscoring the advantage of adapting the transformation to input data distributions.

Notably, the fixed H$_1$ configuration achieves a slightly lower average performance than the baseline. Specifically, the fixed matrix outperforms the baseline on complex tasks such as MNLI, QQP, and QNLI, which suggests that it can suppress saturation to some extent. However, it performs noticeably worse on simpler tasks such as MRPC and RTE. This performance disparity indicates that, although a fixed

Hadamard matrix can partially alleviate saturation, its limited adaptability to varying input distributions may restrict generalization and cause task-specific overfitting.

In summary, these ablation results confirm that each component contributes meaningfully to the overall performance, with learnable parameters providing additional gains through enhanced adaptability to data distributions.

### 5.3. Effect of SmoothSpike

Figure 4 illustrates the distributional shifts of pre-activation values across two representative layers in Spikingformer with SmoothSpike, contrasting states before and after the learnable transformation. The left column depicts pre-transformation distributions, while the right column presents the corresponding post-transformation distributions.

Notably, prior to transformation, pre-activation values exhibit numerous outliers substantially deviating from the normal range, causing neuron saturation. These saturated neurons cannot effectively discriminate among such highly divergent outlier inputs, degrading the model's representational capacity. By contrast, the transformation markedly

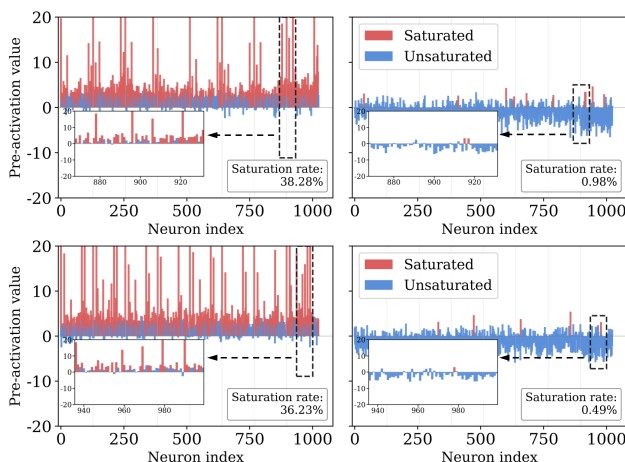

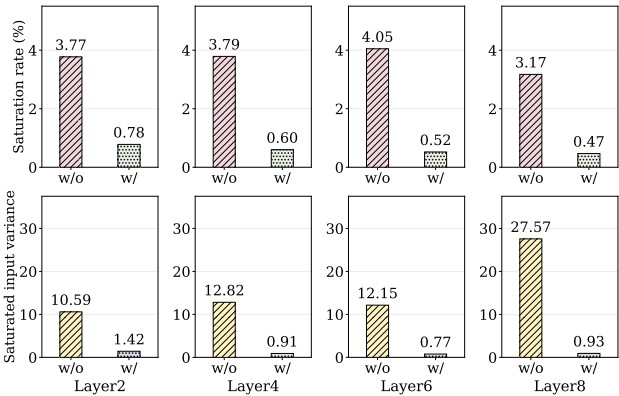

*Figure 4.* Comparison of pre-activation distributions before and after applying the learnable Hadamard transform. The left column shows the distributions from two representative layers before the transform, while the right column shows the distributions after.

*Figure 5.* Comparison of the proportion of saturated neurons (first row) and the input variance of saturated neurons (second row), with (w/) and without (w/o) the learnable Hadamard transform.

smooths these distributions, yielding more uniform distributions (as shown in the right column of Figure 4).

More specifically, as illustrated in Figure 5, we further visualize the saturation rate of all neurons and the variance of inputs to saturated neurons before and after applying the learnable Hadamard transform $\mathbf{H}_3$. After applying $\mathbf{H}_3$, the overall saturation rate is substantially reduced, which means that fewer neurons become saturated and more neurons can preserve input differences in their outputs. Meanwhile, the variance of inputs to saturated neurons also decreases markedly. This suggests that, although a small fraction of neurons may still become saturated, their corresponding input values are much more concentrated. As a result, even when some neurons remain saturated, they collapse a much narrower range of inputs into identical spike outputs than before applying $\mathbf{H}_3$. Overall, these visualization results show that the learnable Hadamard transform effectively reduces information homogenization caused by spike saturation.

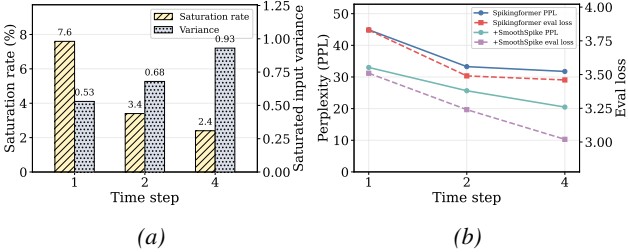

*(a)*                    *(b)*

*Figure 6.* Effect of time steps on spike saturation and performance.

## 5.4. Effect of Time Steps

In this subsection, we study the effect of the simulation time step $T$ using the same 4-layer Spikingformer models as in the ablation experiments. As shown in Fig. 6a, increasing $T$ consistently reduces the saturation rate of the Spikingformer baseline from $7.6\%$ to $3.4\%$ and $2.4\%$ for $T = 1, 2, 4$, respectively. This suggests that a larger temporal budget alleviates widespread neuron saturation. However, the variance of inputs corresponding to still-saturated neurons increases from $0.53$ to $0.68$ and $0.93$, indicating that the remaining saturated cases are dominated by more severe outlier inputs.

Fig. 6b further shows that SmoothSpike improves language modeling performance under all time-step settings. Compared with the baseline, SmoothSpike reduces the validation perplexity on the pre-training corpus by $26.5\%$, $22.9\%$, and $36.0\%$ at $T = 1, 2, 4$, respectively. These results indicate that increasing $T$ and applying SmoothSpike are complementary: the former provides a larger spike budget, while the latter reshapes the pre-activation distribution to mitigate saturation caused by outliers.

## 6. Conclusion

This work identifies spike saturation-induced information homogenization as a critical limitation in Spiking Neural Networks, where bounded temporal windows cause distinct high-amplitude inputs to converge to identical spike counts, compressing representational capacity. To address this issue, we propose SmoothSpike, a novel framework that employs randomized Hadamard transformations to suppress extreme input values, complemented by a learnable orthogonal module adapted via Newton-Schulz iterations to accommodate varying input distributions. We theoretically prove the efficacy of the smoothing mechanism in constraining maximum input magnitudes. Extensive experiments on language modeling tasks demonstrate that SmoothSpike consistently mitigates information homogenization while maintaining the energy efficiency inherent to binary spike-based computation. These results establish that addressing input saturation through orthogonal smoothing offers a principled pathway toward bridging the performance gap between energy-efficient SNNs and conventional ANNs.

## Acknowledgements

This work was supported by the National Natural Science Foundation of China (Grants 62576080 and 62220106008), by the Fundamental and Interdisciplinary Disciplines Breakthrough Plan of the Ministry of Education of China (JYB2025XDXM102), the Guangdong Introducing Innovative and Entrepreneurial Teams (Grant 2023ZT10×044), and the Shenzhen Science and Technology Research Fund (Grant JCYJ20220818103001002).

## Impact Statement

This paper presents work whose goal is to advance the field of Machine Learning. There are many potential societal consequences of our work, none of which we feel must be specifically highlighted here.

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

# A. Proofs of Theorems

## A.1. Proof of Theorem 4.1

By definition, the randomized Hadamard matrix can be written as

$$\mathbf{H}_{\mathrm{R}} = \bar{\mathbf{H}}\mathbf{S}, \tag{16}$$

where $\bar{\mathbf{H}} \in \mathbb{R}^{n \times n}$ is a normalized orthogonal Hadamard matrix with entries $\bar{H}_{ij} \in \{\pm 1/\sqrt{n}\}$, and $\mathbf{S}$ is a random diagonal matrix whose diagonal entries are independent Rademacher random variables uniformly distributed over $\{+1, -1\}$.

**Lemma 1** *(Hoeffding's Inequality) Let $Z_1, Z_2, \ldots, Z_n$ be independent random variables with $Z_i \in [a_i, b_i]$ almost surely. Define $S = \sum_{i=1}^{n} Z_i$. Then for any $t > 0$,*

$$P(|S - \mathbb{E}[S]| \geq t) \leq 2 \exp\left(-\frac{2t^2}{\sum_{i=1}^{n}(b_i - a_i)^2}\right). \tag{17}$$

We prove the concentration bound by first analyzing each coordinate of $\mathbf{H}_{\mathrm{R}}\mathbf{x}$ and then applying a union bound. For a fixed index $i \in \{1, 2, \ldots, n\}$, define

$$Y_i = \mathbf{e}_i^T \mathbf{H}_{\mathrm{R}}\mathbf{x} = \mathbf{e}_i^T \bar{\mathbf{H}}\mathbf{S}\mathbf{x} = \sum_{j=1}^{n} \bar{H}_{ij} x_j S_{jj}. \tag{18}$$

Here, $\bar{H}_{ij}$ and $x_j$ are deterministic, while $S_{jj}$ are independent Rademacher random variables.

Let

$$c_j = \bar{H}_{ij} x_j. \tag{19}$$

Then

$$Y_i = \sum_{j=1}^{n} c_j S_{jj}. \tag{20}$$

Since $\mathbb{E}[S_{jj}] = 0$, we have

$$\mathbb{E}[Y_i] = \sum_{j=1}^{n} c_j \mathbb{E}[S_{jj}] = 0. \tag{21}$$

Each random variable $c_j S_{jj}$ lies in the interval $[-|c_j|, |c_j|]$. Therefore, by Hoeffding's inequality,

$$P(|Y_i| \geq a) \leq 2 \exp\left(-\frac{2a^2}{\sum_{j=1}^{n}(2|c_j|)^2}\right) = 2 \exp\left(-\frac{a^2}{2\sum_{j=1}^{n} c_j^2}\right). \tag{22}$$

Since $\bar{\mathbf{H}}$ is a normalized orthogonal Hadamard matrix, each entry satisfies $\bar{H}_{ij}^2 = 1/n$. Hence,

$$\sum_{j=1}^{n} c_j^2 = \sum_{j=1}^{n} (\bar{H}_{ij} x_j)^2 = \frac{1}{n} \sum_{j=1}^{n} x_j^2 = \frac{\|\mathbf{x}\|_2^2}{n}. \tag{23}$$

Substituting this into the previous bound gives

$$P(|Y_i| \geq a) \leq 2 \exp\left(-\frac{na^2}{2\|\mathbf{x}\|_2^2}\right). \tag{24}$$

Applying the union bound over all coordinates, we obtain

$$P\left(\max_{i \in \{1, \ldots, n\}} |Y_i| \geq a\right) \leq \sum_{i=1}^{n} P(|Y_i| \geq a) \leq 2n \exp\left(-\frac{na^2}{2\|\mathbf{x}\|_2^2}\right). \tag{25}$$

Choosing

$$a = \|\mathbf{x}\|_2 \sqrt{\frac{2}{n} \log\left(\frac{2n}{\epsilon}\right)} \tag{26}$$

yields

$$2n \exp\left(-\frac{na^2}{2\|\mathbf{x}\|_2^2}\right) = \epsilon. \tag{27}$$

Therefore,

$$P\left(\max_i |\mathbf{e}_i^T \mathbf{H}_{\mathrm{R}} \mathbf{x}| \geq \|\mathbf{x}\|_2 \sqrt{\frac{2}{n} \log\left(\frac{2n}{\epsilon}\right)}\right) \leq \epsilon. \tag{28}$$

This completes the proof.

### A.2. Proof of Theorem 4.2

Let $\widetilde{\mathbf{H}} \in \mathbb{R}^{n \times n}$ be a full-rank matrix with singular value decomposition:

$$\widetilde{\mathbf{H}} = \mathbf{U}\mathbf{\Sigma}\mathbf{V}^\top \tag{29}$$

where $\mathbf{U}, \mathbf{V} \in \mathcal{O}(n)$ are orthogonal matrices and $\mathbf{\Sigma} = \mathrm{diag}(\sigma_1, \sigma_2, \ldots, \sigma_n)$ is a diagonal matrix with $\sigma_i > 0$ for all $i$.

For any orthogonal matrix $\mathbf{H}' \in \mathcal{O}(n)$, we minimize:

$$\|\widetilde{\mathbf{H}} - \mathbf{H}'\|_F^2 = \mathrm{tr}[(\widetilde{\mathbf{H}} - \mathbf{H}')^\top (\widetilde{\mathbf{H}} - \mathbf{H}')] \tag{30}$$

Expanding the trace expression yields:

$$\|\widetilde{\mathbf{H}} - \mathbf{H}'\|_F^2 = \mathrm{tr}(\widetilde{\mathbf{H}}^\top \widetilde{\mathbf{H}}) - 2\mathrm{tr}(\widetilde{\mathbf{H}}^\top \mathbf{H}') + \mathrm{tr}(\mathbf{H}'^\top \mathbf{H}') \tag{31}$$

Since $\mathbf{H}'$ is orthogonal, we have $\mathrm{tr}(\mathbf{H}'^\top \mathbf{H}') = n$. Moreover, $\mathrm{tr}(\widetilde{\mathbf{H}}^\top \widetilde{\mathbf{H}}) = \sum_{i=1}^n \sigma_i^2$ is constant with respect to $\mathbf{H}'$. Consequently, the minimization problem is equivalent to:

$$\max_{\mathbf{H}' \in \mathcal{O}(n)} \mathrm{tr}(\widetilde{\mathbf{H}}^\top \mathbf{H}') \tag{32}$$

Substituting the singular value decomposition of $\widetilde{\mathbf{H}}$:

$$\mathrm{tr}(\widetilde{\mathbf{H}}^\top \mathbf{H}') = \mathrm{tr}(\mathbf{V}\mathbf{\Sigma}\mathbf{U}^\top \mathbf{H}') \tag{33}$$

By the cyclic property of the trace operator:

$$\mathrm{tr}(\widetilde{\mathbf{H}}^\top \mathbf{H}') = \mathrm{tr}(\mathbf{\Sigma}\mathbf{U}^\top \mathbf{H}'\mathbf{V}) \tag{34}$$

Define $\mathbf{R} = \mathbf{U}^\top \mathbf{H}'\mathbf{V}$. Since $\mathbf{U}, \mathbf{H}', \mathbf{V}$ are all orthogonal matrices, $\mathbf{R} \in \mathcal{O}(n)$. Thus:

$$\mathrm{tr}(\widetilde{\mathbf{H}}^\top \mathbf{H}') = \mathrm{tr}(\mathbf{\Sigma}\mathbf{R}) = \sum_{i=1}^n \sigma_i r_{ii} \tag{35}$$

where $r_{ii}$ denotes the $i$-th diagonal element of $\mathbf{R}$.

For any orthogonal matrix $\mathbf{R}$, the diagonal elements satisfy $|r_{ii}| \leq 1$ for all $i$. Since $\sigma_i > 0$, the sum $\sum_{i=1}^n \sigma_i r_{ii}$ is maximized when $r_{ii} = 1$ for all $i$, which corresponds to $\mathbf{R} = \mathbf{I}$. The maximum value is:

$$\max_{\mathbf{R} \in \mathcal{O}(n)} \mathrm{tr}(\mathbf{\Sigma}\mathbf{R}) = \sum_{i=1}^n \sigma_i \tag{36}$$

The condition $\mathbf{R} = \mathbf{I}$ implies:

$$\mathbf{U}^\top \mathbf{H}'\mathbf{V} = \mathbf{I} \quad \Rightarrow \quad \mathbf{H}' = \mathbf{U}\mathbf{V}^\top \tag{37}$$

This uniquely determines the optimal orthogonal approximation as $\mathbf{H} = \mathbf{U}\mathbf{V}^\top = \mathrm{msign}(\widetilde{\mathbf{H}})$.

Now consider the case where $\widetilde{\mathbf{H}}$ is rank-deficient with $\mathrm{rank}(\widetilde{\mathbf{H}}) = r < n$. The singular value decomposition takes the form:

$$\widetilde{\mathbf{H}} = \mathbf{U}\mathbf{\Sigma}\mathbf{V}^\top \tag{38}$$

where $\mathbf{\Sigma} = \mathrm{diag}(\sigma_1, \ldots, \sigma_r, 0, \ldots, 0)$ with $\sigma_i > 0$ for $i \leq r$ and $\sigma_i = 0$ for $i > r$. The optimization problem remains:

$$\max_{\mathbf{H}' \in \mathcal{O}(n)} \mathrm{tr}(\widetilde{\mathbf{H}}^\top \mathbf{H}') = \max_{\mathbf{R} \in \mathcal{O}(n)} \mathrm{tr}(\mathbf{\Sigma}\mathbf{R}) = \max_{\mathbf{R} \in \mathcal{O}(n)} \sum_{i=1}^{r} \sigma_i r_{ii} \tag{39}$$

Since only the first $r$ singular values are nonzero, the maximum is attained when $r_{ii} = 1$ for $i = 1, \ldots, r$. The remaining diagonal elements $r_{ii}$ for $i > r$ can take any values satisfying the orthogonality constraints of $\mathbf{R}$. Therefore, the choice $\mathbf{R} = \mathbf{I}$ remains feasible and achieves the maximum:

$$\max_{\mathbf{R} \in \mathcal{O}(n)} \sum_{i=1}^{r} \sigma_i r_{ii} = \sum_{i=1}^{r} \sigma_i \tag{40}$$

This corresponds to $\mathbf{H}' = \mathbf{U}\mathbf{V}^\top$, which is precisely $\mathrm{msign}(\widetilde{\mathbf{H}})$ as defined for rank-deficient matrices.

In both the full-rank and rank-deficient cases, the matrix sign function achieves the minimum Frobenius distance to $\widetilde{\mathbf{H}}$ over all orthogonal matrices:

$$\mathbf{H} = \mathrm{msign}(\widetilde{\mathbf{H}}) = \arg\min_{\mathbf{H}' \in \mathcal{O}(n)} \|\widetilde{\mathbf{H}} - \mathbf{H}'\|_F$$

This completes the proof.

## B. Approximation of the Msign Function Using Newton-Schulz Iteration

The Newton-Schulz iteration provides an iterative method to approximate the matrix sign function, $\mathrm{msign}(\widetilde{\mathbf{H}})$, for a matrix $\widetilde{\mathbf{H}} \in \mathbb{R}^{m \times n}$ with singular value decomposition $\widetilde{\mathbf{H}} = \mathbf{U}\mathbf{\Sigma}\mathbf{V}^\top$. As defined, $\mathrm{msign}(\widetilde{\mathbf{H}}) = \mathbf{U}\mathbf{V}^\top$, which retains the left and right singular vectors while replacing all singular values with 1.

To apply the iteration, first normalize $\widetilde{\mathbf{H}}$ to obtain $\mathbf{Y}_0 = \widetilde{\mathbf{H}}/\|\widetilde{\mathbf{H}}\|_{\ell_2 \to \ell_2}$ (using the spectral norm, ensuring all singular values of $\mathbf{Y}_0$ lie in $(0, 1]$) or alternatively $\mathbf{Y}_0 = \widetilde{\mathbf{H}}/\|\widetilde{\mathbf{H}}\|_F$ (using the Frobenius norm). Then, iterate:

$$\mathbf{Y}_{t+1} = \frac{3}{2}\mathbf{Y}_t - \frac{1}{2}\mathbf{Y}_t(\mathbf{Y}_t^\top \mathbf{Y}_t). \tag{41}$$

As $t \to \infty$, $\mathbf{Y}_t \to \mathbf{U}\mathbf{V}^\top = \mathrm{msign}(\widetilde{\mathbf{H}})$, provided the singular values of $\mathbf{Y}_0$ are in $(0, \sqrt{3})$.

This convergence arises because the iteration effectively applies the univariate cubic function $f(x) = \frac{3}{2}x - \frac{1}{2}x^3$ to each singular value of $\mathbf{Y}_t$. For $0 < x < \sqrt{3}$, repeated application of $f(x)$ drives the value toward 1, approximating the sign function $\mathrm{sign}(x)$ (which is 1 for positive $x$). The spectral normalization ensures the initial singular values satisfy this condition more robustly than the Frobenius norm, though the latter may suffice in practice.

More generally, the Newton-Schulz iteration belongs to a family of degree-$2n + 1$ polynomial iterations of the form

$$\mathbf{Y}_{t+1} = a\mathbf{Y}_t + b\mathbf{Y}_t(\mathbf{Y}_t^\top \mathbf{Y}_t) + c\mathbf{Y}_t(\mathbf{Y}_t^\top \mathbf{Y}_t)^2 + \cdots + z\mathbf{Y}_t(\mathbf{Y}_t^\top \mathbf{Y}_t)^n, \tag{42}$$

where coefficients $a, b, c, \ldots, z$ are chosen such that the corresponding univariate polynomial $g(x) = ax + bx^3 + cx^5 + \cdots + zx^{2n+1}$ approximates $\mathrm{sign}(x)$. These coefficients can be tuned to optimize convergence speed. This approach is classical, dating back to works such as (Kovarik, 1970) and (Björck & Bowie, 1971), and is discussed in (Higham, 2008).

In our experiments, we adopt a quintic Newton-Schulz-style iteration of the form

$$\mathbf{Y}_{t+1} = a\mathbf{Y}_t + b\,\mathbf{Y}_t(\mathbf{Y}_t^\top \mathbf{Y}_t) + c\,\mathbf{Y}_t(\mathbf{Y}_t^\top \mathbf{Y}_t)^2, \tag{43}$$

with empirically tuned coefficients $a = 3.4445$, $b = -4.7750$, and $c = 2.0315$. These coefficients are chosen to achieve faster per-iteration convergence at the expense of strict guaranteed convergence to $\mathrm{msign}(\widetilde{\mathbf{H}})$, trading some approximation accuracy for computational efficiency in large-scale deep learning settings.

For backpropagation, given the gradient of the loss $\mathcal{L}$ with respect to $\mathbf{Y}_k$ as $\partial\mathcal{L}/\partial\mathbf{Y}_k$, gradients propagate to $\widetilde{\mathbf{H}}$ via the chain rule. The backpropagation for the $k$-th step is:

$$
\begin{aligned}
\frac{\partial\mathcal{L}}{\partial\mathbf{Y}_k} &= a\frac{\partial\mathcal{L}}{\partial\mathbf{Y}_{k+1}} + b\Big[\frac{\partial\mathcal{L}}{\partial\mathbf{Y}_{k+1}}(\mathbf{Y}_k^\top\mathbf{Y}_k) + \mathbf{Y}_k\Big(\frac{\partial\mathcal{L}}{\partial\mathbf{Y}_{k+1}}^\top\mathbf{Y}_k + \mathbf{Y}_k^\top\frac{\partial\mathcal{L}}{\partial\mathbf{Y}_{k+1}}\Big)\Big] \\
&+ c\Big[\frac{\partial\mathcal{L}}{\partial\mathbf{Y}_{k+1}}(\mathbf{Y}_k^\top\mathbf{Y}_k)^2 + 2\mathbf{Y}_k(\mathbf{Y}_k^\top\mathbf{Y}_k)\Big(\frac{\partial\mathcal{L}}{\partial\mathbf{Y}_{k+1}}^\top\mathbf{Y}_k + \mathbf{Y}_k^\top\frac{\partial\mathcal{L}}{\partial\mathbf{Y}_{k+1}}\Big)\Big].
\end{aligned}
\tag{44}
$$

## C. Fusion of the RMSNorm Scale into Spiking Thresholds

RMSNorm applies a channel-wise learnable scale after normalizing the input by its root-mean-square value. Given an input $\mathbf{x}^t \in \mathbb{R}^n$ at time step $t$, it can be written as

$$
\mathrm{RN}(\mathbf{x}^t) = \boldsymbol{\gamma}\odot\bar{\mathbf{x}}^t, \qquad \bar{\mathbf{x}}^t = \frac{\mathbf{x}^t}{\sqrt{\frac{1}{n}\|\mathbf{x}^t\|_2^2 + \epsilon}},
\tag{45}
$$

where $\boldsymbol{\gamma} \in \mathbb{R}^n$ is the learnable channel-wise scaling parameter and $\epsilon$ is a small constant for numerical stability.

When RMSNorm is directly followed by spiking neurons, the scale parameter $\boldsymbol{\gamma}$ can be equivalently absorbed into the firing thresholds. Consider the $i$-th channel of a standard spiking neuron with threshold $\vartheta_i$. For clarity, we use the following leaky integrate-and-fire update with subtractive reset:

$$
u_i^t = \lambda u_i^{t-1} + \gamma_i\bar{x}_i^t - \vartheta_i s_i^{t-1}, \qquad s_i^t = \mathbb{I}\left(u_i^t \geq \vartheta_i\right),
\tag{46}
$$

where $u_i^t$ denotes the membrane potential, $s_i^t$ is the output spike, and $\lambda$ is the leak factor. Since $\gamma_i$ acts as a positive channel-wise gain, we define a rescaled membrane potential and threshold as

$$
\tilde{u}_i^t = \frac{u_i^t}{\gamma_i}, \qquad \tilde{\vartheta}_i = \frac{\vartheta_i}{\gamma_i}.
\tag{47}
$$

Dividing the membrane update by $\gamma_i$ gives

$$
\tilde{u}_i^t = \lambda\tilde{u}_i^{t-1} + \bar{x}_i^t - \tilde{\vartheta}_i s_i^{t-1}, \qquad s_i^t = \mathbb{I}\left(\tilde{u}_i^t \geq \tilde{\vartheta}_i\right).
\tag{48}
$$

Thus, scaling the RMS-normalized input by $\gamma_i$ is exactly equivalent to using the unscaled RMS-normalized input with a rescaled threshold $\tilde{\vartheta}_i = \vartheta_i/\gamma_i$. Therefore, the spike train remains unchanged after the fusion.

## D. Datasets and Experimental Settings

### D.1. Datasets

**Corpus of Linguistic Acceptability (CoLA)** The Corpus of Linguistic Acceptability (Warstadt et al., 2019) consists of English sentences extracted from linguistic literature, annotated by linguists for grammatical acceptability. The task requires binary classification of whether a given sentence conforms to the grammatical constraints of standard English, serving as a fundamental assessment of a model's syntactic competence. Performance is evaluated using both Matthews Correlation Coefficient (MCC) and accuracy.

**Stanford Sentiment Treebank (SST-2)** Derived from movie reviews with manually annotated parse trees, SST-2 (Socher et al., 2013) provides a benchmark for sentiment classification at the sentence level. Each instance comprises a sentence annotated with binary sentiment polarity (positive or negative). The dataset challenges models to capture compositional sentiment semantics without utilizing structural phrase-level annotations during fine-tuning.

**Microsoft Research Paraphrase Corpus (MRPC)** MRPC (Dolan & Brockett, 2005) comprises sentence pairs automatically extracted from online news sources, with human annotations indicating whether the pairs are semantically equivalent (paraphrases). This binary classification task evaluates the model's ability to recognize semantic equivalence despite lexical and syntactic variation between source and target sentences.

**Semantic Textual Similarity Benchmark (STS-B)**    Sourced from news headlines, video and image captions, and natural language inference datasets, STS-B (Cer et al., 2017) requires the prediction of continuous similarity scores ranging from 0 to 5 for sentence pairs. The task assesses the model's capacity to quantify semantic relatedness, with performance measured by Pearson and Spearman correlation coefficients against human judgments.

**Quora Question Pairs (QQP)**    This dataset consists of question pairs derived from the Quora platform, annotated to identify duplicate questions that seek identical information. The binary classification task tests the model's ability to distinguish between semantically equivalent inquiries despite surface form variations, presenting challenges in paraphrase detection and semantic matching.

**Multi-Genre Natural Language Inference (MNLI)**    MNLI (Williams et al., 2018) expands natural language inference across multiple genres of written and spoken English, including transcribed speech, popular fiction, and government reports. Given a premise and hypothesis sentence, the task requires three-way classification into entailment, contradiction, or neutral relationships. The dataset includes matched (in-domain) and mismatched (cross-domain) test sets to evaluate generalization capabilities.

**Question-answering Natural Language Inference (QNLI)**    Constructed from the Stanford Question Answering Dataset (Rajpurkar et al., 2016), QNLI presents a sentence-level entailment task where premise sentences (contextual paragraphs) and hypothesis sentences (questions) are paired. The binary classification task determines whether the premise sentence contains the answer to the question, effectively converting reading comprehension into textual entailment.

**Recognizing Textual Entailment (RTE)**    RTE aggregates data from annual textual entailment challenges (RTE1–RTE5) (Dagan et al., 2005; Haim et al., 2006; Giampiccolo et al., 2007; Bentivogli et al., 2009), combining news and Wikipedia text into a binary classification benchmark. The task requires the model to identify whether a given hypothesis can be inferred from the corresponding premise, representing a fundamental evaluation of natural language inference capabilities.

### D.2. Experimental Settings

Following the standard BERT paradigm, we implement Spikingformer with SmoothSpike and SpikeLM with SmoothSpike via pre-training and fine-tuning.

**Pre-training Configuration**    We perform masked language modeling (MLM) pre-training with a masking probability of 15%. We use a global batch size of 512, distributed across 8 nodes with 64 sequences per device and a gradient accumulation step of 1. Optimization is performed using AdamW with an initial learning rate of $2 \times 10^{-4}$ and no weight decay. The learning rate follows a linear decay schedule after a linear warm-up of 5,000 steps. Each sequence contains a maximum of 128 tokens. Spikingformer with SmoothSpike is trained on approximately 36.0 billion tokens, compared to 52.4 billion for SpikeLM with SmoothSpike. All datasets are accessed via the Hugging Face Datasets library: Stories[3], BookCorpus[4], CC-News[5], OpenWebText[6], and Wikipedia[7].

**Fine-tuning Configuration**    We fine-tune the pre-trained model on individual GLUE tasks, initializing from the checkpoint. For Spikingformer with SmoothSpike, we employ a per-device batch size of 32, whereas for SpikeLM with SmoothSpike, the batch size is set to 16. The maximum sequence length is 128 tokens. We use a learning rate of $2 \times 10^{-5}$ without weight decay, following a linear decay schedule without warm-up steps. Gradient accumulation is disabled.

**Evaluation Metrics**    For QQP, we use the evaluation metric reported by each corresponding baseline paper for fair comparison. The original Spikingformer paper reports accuracy on QQP, while the original SpikeLM paper reports F1. Accordingly, we report accuracy for Spikingformer+SmoothSpike and F1 for SpikeLM+SmoothSpike on QQP. For the other

---

[3]https://huggingface.co/datasets/roneneldan/TinyStories
[4]https://huggingface.co/datasets/bookcorpus
[5]https://huggingface.co/datasets/vblagoje/cc_news
[6]https://huggingface.co/datasets/Skylion007/openwebtext
[7]https://huggingface.co/datasets/wikimedia/wikipedia

GLUE tasks, we report F1 for MRPC, Spearman correlation for STS-B, Matthews correlation coefficient for CoLA, and accuracy for the remaining tasks.

## E. Summary of Notation

Throughout this paper, we use bold uppercase letters to denote matrices and bold lowercase letters to denote vectors. The main notation used in our SmoothSpike framework is summarized in Table 3.

*Table 3.* Summary of notation used in SmoothSpike.

| Notation | Description |
|---|---|
| $\mathbf{x}^\ell \in \mathbb{R}^n$ | Pre-activation input of spiking neurons at layer $\ell$ |
| $\widetilde{\mathbf{x}}^\ell$ | Transformed pre-activation input after orthogonal smoothing |
| $\mathbf{s}^\ell$ | Output spike train generated by spiking neurons |
| $\mathbf{y}^\ell$ | Decoded output representation after the inverse transform |
| $T$ | Length of the simulation time window |
| $n$ | Hidden dimension or feature dimension |
| $\mathbf{H}_n$ | Unnormalized Hadamard matrix with entries in $\{-1, +1\}$ |
| $\bar{\mathbf{H}} = \frac{1}{\sqrt{n}}\mathbf{H}_n$ | Normalized Hadamard matrix satisfying $\bar{\mathbf{H}}^\top \bar{\mathbf{H}} = \mathbf{I}$ |
| $\mathbf{S} = \mathrm{diag}(\sigma_1, \ldots, \sigma_n)$ | Random diagonal matrix with independent Rademacher entries |
| $\mathbf{H}_\mathrm{R} = \bar{\mathbf{H}}\mathbf{S}$ | Randomized Hadamard matrix used for input smoothing |
| $\widetilde{\mathbf{H}}$ | Learnable unconstrained transformation matrix |
| $\mathbf{H} = \mathrm{msign}(\widetilde{\mathbf{H}})$ | Orthogonal matrix obtained by projecting $\widetilde{\mathbf{H}}$ via the matrix sign function |
| $\mathbf{H}_p^\ell$ | Orthogonal transform at layer $\ell$ and transform position $p$ |
| $\mathrm{SN}(\cdot)$ | Spiking neuron operation within a window of length $T$ |
| $\mathrm{RN}(\cdot)$ | Root mean square layer normalization with learnable scaling |
| $\mathrm{RN}_0(\cdot)$ | Scale-free RMSNorm after absorbing the learnable scaling into firing thresholds |
| $\mathbf{W}^\ell$ | Weight matrix of a linear layer at layer $\ell$ |
| $\widetilde{\mathbf{W}}^\ell$ | Fused weight matrix after absorbing the orthogonal transform |
| $\vartheta$ | Firing threshold of spiking neurons |

