# OpenReview forum: "SmoothSpike: Spiking Transformer with Learnable Hadamard Transformation"
_ICML.cc/2026/Conference — ICML 2026 spotlight_

### Official Review · Reviewer_RCMA · 2026-03-09

**Soundness:** 3
**Presentation:** 3
**Significance:** 3
**Originality:** 3
**Overall Recommendation:** 5
**Confidence:** 4

**Summary:**

This paper introduces SmoothSpike, a method to enhance the representational capacity of Spiking Neural Networks (SNNs) in language modeling by addressing spike saturation-induced information homogenization. It proposes a learnable orthogonal transformation initialized with randomized Hadamard matrices, theoretically proven to suppress extreme inputs and reduce saturation. Experiments on the GLUE benchmark demonstrate performance improvements.

**Compliance With Llm Reviewing Policy:**

Affirmed.

**Final Justification:**

The author provided a strong response to my question, and I am willing to increase my score.

**Key Questions For Authors:**

1. Why was the Hadamard transformation not applied to query/key neurons in attention mechanisms, and what ablation results support this decision?
2. How does SmoothSpike perform on non-language tasks, such as image classification?
3. The theorem on maximum element bounds assumes a randomized sign matrix; how sensitive is the method to different randomization strategies, and are there empirical robustness tests?
4. Given that standard Hadamard matrices strictly require dimensions of $n=2^k$, how do the authors construct $H_1$ and $H_2$ for the 768-dimensional hidden states ($768=2^8 \times 3$) employed in the paper?
5. Does the 5th-order Newton-Schulz iteration maintain stable convergence across all layers and training stages? Can convergence issues lead to violations of the orthogonal constraint?

**Limitations:**

Yes

**Strengths And Weaknesses:**

Strengths:
1. This paper identifies saturation-induced information homogenization as a critical limitation in SNNs. Then, it proposes Hadamard transformations as a solution, supported by both theoretical guarantees and empirical evidence.
2. The learnable orthogonal matrix approach, approximated via Newton-Schulz iteration, allows adaptive smoothing while maintaining orthogonality for efficient inference fusion, showing substantial gains over baselines.

Weaknesses:
1. The evaluation is limited to the GLUE benchmark and specific baselines, lacking comparisons on diverse datasets like vision tasks.
2. The block-diagonal constraint on certain matrices could introduce unknown biases.
3. The ablation studies lack an analysis of interactions with other SNN enhancements, such as multi-level spikes.

---

> ### Author Rebuttal · Authors · 2026-03-29
>
> We are deeply grateful for your time in reviewing our manuscript. Your insights will be of great help to us in improving the quality of our paper.
>
> ## Response to Weakness 1 & Question 2:
>
> We acknowledge that current experiments center on GLUE. During rebuttal, we validated SmoothSpike on vision tasks. Saturation analysis on a vision Spikingformer shows the issue persists (up to ~6% in some layers), confirming that saturation stems from the fundamental constraint of spike representation under finite time windows, not task-specific properties. Fine-tuning the pretrained Spikingformer-8-768 with SmoothSpike for 50 epochs yields:
>
> |Method|Acc.|
> |-|-|
> |Spikingformer-8-768|75.85|
> |+SmoothSpike|**76.98**|
>
> SmoothSpike operates on the pre-activation distribution and requires no task-specific architectural modifications—only inserting orthogonal transforms before/after saturation-prone spiking neurons.
>
> ## Response to Weakness 2:
>
> The block-diagonal constraint on $H_3$ is a computational efficiency design, not a semantic inductive bias. Each block still performs a full orthogonal transform for energy redistribution. Moreover, the fully-connected linear layers before and after SmoothSpike can mix information across blocks, so the constraint only applies locally within the smoothing step.
>
> Empirically, adding block-diagonal $H_3$ improves average performance (67.64→67.88 in Table 2), showing no systematic harm. A direct comparison confirms the four-block setting incurs minimal degradation versus a full orthogonal matrix while being **1.5×** faster:
>
> |Method|perplexity (↓)|eval loss (↓)|
> |-|-|-|
> |Full|20.31|3.01|
> |4 blocks|20.49|3.02|
>
> ## Response to Weakness 3:
>
> Our SpikeLM experiments already address this: SpikeLM uses bidirectional/ternary-style spikes, so improvements atop it demonstrate compatibility with multi-level spike mechanisms. SmoothSpike and multi-level spikes are complementary—the latter expands discrete output sets, while SmoothSpike reshapes the pre-activation distribution to reduce saturation risk, which persists even with richer spike levels under finite $T$.
>
> |Model|T|Energy|MNLI$_{-m/mm}$|QQP|QNLI|SST-2|CoLA|STS-B|MRPC|RTE|Avg.|
> |-|-|-|-|-|-|-|-|-|-|-|-|
> |SpikeLM|1|3.98|76.0/76.9|84.0|84.9|86.5|37.9|84.3|85.6|65.3|75.7|
> |SpikeLM|4|13.74|77.1/77.2|83.9|85.3|87.0|38.8|84.9|85.7|69.0|76.5|
> |+SmoothSpike|1|7.03|76.8/77.7|84.3|86.8|89.5|52.7|83.5|88.1|58.1|**77.5**|
>
> Notably, SmoothSpike at $T{=}1$ outperforms SpikeLM at $T{=}4$ by 1.0 point while using 48.8% less energy. For integer-valued neurons, we believe compatibility holds in principle but defer systematic validation to future work as joint training stability needs further tuning.
>
> ## Response to Question 1:
>
> Empirically, Q/K neurons exhibit only mild saturation, so adding transforms there is not cost-effective. Our ablation confirms that introducing $H_{Q,K}$ causes slight degradation:
>
> |Method|perplexity (↓)|eval loss (↓)|
> |-|-|-|
> |$H_1$,$H_2$|**22.00**|**3.09**|
> |$H_1$,$H_2$,$H_{Q,K}$|22.43|3.11|
>
> ## Response to Question 3:
>
> We compared random orthogonal matrices vs. random Hadamard initialization:
>
> |Randomization Strategy|perplexity (↓)|eval loss (↓)|
> |-|-|-|
> |Random Orthogonal Matrix|22.10|3.10|
> |Random Hadamard Matrix|**20.49**|**3.02**|
>
> Both work well, confirming robustness to randomization family. Random Hadamard is superior because every element has magnitude $1/\sqrt{n}$, providing stronger uniform mixing and enabling the explicit bound in Theorem 4.1.
>
> ## Response to Question 4:
>
> In our implementation, we adopt the `fast-hadamard-transform` library (Dao, 2024), which provides a CUDA-optimized fast Hadamard transform with a PyTorch interface. This library natively supports dimensions that are not powers of two: when the input dimension is not $2^k$, the kernel implicitly zero-pads the input to the next power of two internally and returns an output of the same dimension as the input. Therefore, for the 768-dimensional hidden states used in our experiments, we simply invoke `hadamard_transform(x)` directly without any manual padding or truncation. The entire non-power-of-two handling is performed transparently within the CUDA kernel. We will clarify this implementation detail in the revised manuscript.
>
> ## Response to Question 5:
>
> The 5th-order Newton-Schulz iteration is a practical approximation. Its stability derives from: (1) Hadamard initialization places singular values near 1; (2) forward-pass matrices are continuously projected toward orthogonality; (3) the sandwiched $H/H^T$ form prevents error accumulation.
>
> We measured orthogonality deviation $\delta_{\text{orth}}(H)=\lVert H^TH-I\rVert_F/\lVert I\rVert_F$ across all layers: $H_1$: 0.27, $H_2$: 0.28–0.43, $H_3$: 0.26–0.31—small and stable across depth. Furthermore, inference performance on GLUE is identical before and after weight fusion, confirming that any residual deviation has no practical impact.
>
> We will revise our paper based on your feedback, and we thank you again.

---

### Official Review · Reviewer_DwvF · 2026-03-11

**Soundness:** 3
**Presentation:** 3
**Significance:** 2
**Originality:** 2
**Overall Recommendation:** 4
**Confidence:** 5

**Summary:**

- This paper studies a representational bottleneck in spiking Transformers for language tasks, which the authors describe as spike saturation-induced information homogenization.
- The core argument is that under a bounded time window, distinct high-amplitude inputs can collapse to the same maximal spike count, thereby reducing discriminability in spiking language models.
- To mitigate this issue, the paper proposes SmoothSpike, which inserts an orthogonal transformation before spiking neurons to smooth extreme pre-activation values, followed by an inverse transformation after spike generation.
- The method is motivated first by a randomized Hadamard transform, and then extended to a learnable orthogonal transform that is approximately maintained during training using a Newton–Schulz-style update.
- The paper provides theoretical analysis showing that the randomized Hadamard transform can bound the maximum coordinate magnitude and thus reduce saturation risk, and also discusses the orthogonal projection property of the matrix-sign operation.
- Empirically, SmoothSpike is integrated into Spikingformer and SpikeLM and evaluated on GLUE, where it consistently improves performance, with especially large gains for Spikingformer.
- Overall, the paper presents a clean and meaningful idea: improving spiking language models by reducing saturation-induced representational collapse through orthogonal smoothing.

**Compliance With Llm Reviewing Policy:**

Affirmed.

**Ethical Review Concerns:**

.

**Final Justification:**

The authors provided a strong and constructive rebuttal. My main concerns were adequately addressed, and the clarification improved my confidence in the paper. Accordingly, I am willing to increase my score.

**Key Questions For Authors:**

- The theoretical analysis shows that randomized Hadamard smoothing reduces extreme coordinate magnitudes. Can the authors provide stronger evidence that this translates into improved task-relevant information retention, rather than simply smoother activation statistics?
- Since the learnable orthogonal transform is the practically important version, can the authors quantify how close the learned matrices remain to exact orthogonality during training, and how sensitive performance is to that approximation quality?
- The paper evaluates only on GLUE. Do the authors expect the same saturation/homogenization mechanism to matter in autoregressive or generative language modeling, where activation statistics and error modes may differ substantially?
- Table 1 suggests that SmoothSpike improves task performance but increases reported energy for both backbones. Can the authors clarify whether the main intended contribution is better accuracy under SNN constraints, or an improved accuracy-efficiency trade-off?
- The fixed Hadamard transform sometimes performs worse than the baseline, while the learnable version helps. Can the authors provide more insight into what structure the learned transforms capture beyond generic rotation/smoothing?

**Limitations:**

The paper does acknowledge future directions, including hardware validation and larger-scale pretraining settings, which implicitly recognize meaningful current limitations.

**Strengths And Weaknesses:**

### Strengths
- The paper identifies a concrete and plausible failure mode in spiking language models. The saturation/homogenization framing is much more specific and insightful than a generic claim that SNNs have limited expressivity.
- The proposed method is conceptually clean. Using an orthogonal transform to redistribute activation mass before spike generation is a neat intervention, and the inverse-transform design keeps the method structurally coherent.
- The paper provides meaningful theoretical support for the core idea. In particular, the randomized Hadamard analysis gives a non-trivial justification for why smoothing can reduce the risk of coordinate-wise saturation. This does not fully prove downstream task improvement, but it does make the method scientifically motivated rather than purely heuristic.
- The transition from a fixed randomized transform to a learnable orthogonal transform is practically sensible. It preserves the core motivation while allowing the model to adapt the smoothing operation to the task.
- The empirical results are strong enough to matter. The gains on GLUE, especially for Spikingformer, are substantial and suggest that the proposed mechanism is not merely a cosmetic modification.
- The ablation studies are useful. The paper shows that adding multiple smoothing locations helps, and that the learnable version performs better than the fixed transform.
- The diagnostic analyses are aligned with the claimed mechanism. The activation histograms and saturation statistics provide supporting evidence that SmoothSpike is actually changing the pre-activation distribution in the intended way.
- The paper is generally well organized and readable. The motivation, method, theory, and empirical evidence connect reasonably well.

### Weaknesses
- While the theory is meaningful, it still supports a narrower claim than the full empirical narrative. The main theorem shows that randomized Hadamard smoothing reduces extreme coordinate magnitudes, but it does not directly prove that the trained model preserves more task-relevant semantic information or that homogenization is the dominant cause of downstream errors.
- The practical method relies on a learnable orthogonal transform, whereas the strongest theoretical result is derived for the fixed randomized Hadamard case. This creates some gap between the most rigorous theoretical object and the version that actually delivers the main empirical gains.
- The learnable orthogonality maintenance is only approximate. The Newton–Schulz-style update is practical, but not fully satisfying from a rigorous standpoint, especially since the paper acknowledges a trade-off against strict convergence guarantees.
- The evaluation is somewhat narrow relative to the broad framing of the paper. The experiments are limited to GLUE-style sentence understanding tasks, so it remains unclear how well the proposed mechanism transfers to autoregressive language modeling, long-context settings, or larger-scale LLM-style workloads.
- SmoothSpike improves accuracy, but it also increases reported energy relative to the base SNN backbones. So the contribution is better viewed as improved representational quality within the SNN framework rather than a strict Pareto improvement in accuracy and efficiency.

---

> ### Author Rebuttal · Authors · 2026-03-29
>
> We are deeply grateful for your time in reviewing our manuscript. Your insights will be of great help to us in improving the quality of our paper.
> ## Response to Weakness 1:
> We agree that Theorem 4.1 establishes a mechanism-level result—suppressing extreme coordinates to reduce saturation—rather than a complete theory of semantic preservation.
>
> The key evidence beyond "smoother statistics" lies in two quantities: $p_{\text{sat}}=\Pr(c(x)=T)$ and $\Delta_{\text{sat}}=\mathrm{Var}(x \mid c(x)=T)$. Figure 4 shows both decrease substantially after applying SmoothSpike, meaning fewer samples are compressed to the maximum spike count $T$, and even when saturation occurs, the merged input interval is narrower. Combined with the 8.2-point average improvement (Table 1), these results support that reducing homogenization preserves task-relevant differences. We will reframe homogenization as one key bottleneck rather than the sole cause.
> ## Response to Weakness 2:
> Theorem 4.1 validates the core smoothing mechanism: orthogonal transforms preserve $\lVert x\rVert_2$ and redistribute energy, with randomized Hadamard providing an explicit $\ell_\infty$ bound. The learnable version is a data-adaptive instantiation within the same orthogonal framework, using $Q=\mathrm{msign}(W)\in\mathcal{O}(n)$, which by Theorem 4.3 is the optimal orthogonal approximation of $W$. Supporting ablation results:
> |Initialization|Perplexity↓|Eval Loss↓|
> |-|-|-|
> |Base|31.75|3.46|
> |Identity|28.40|3.35|
> |Random Orthogonal|22.10|3.10|
> |Random Hadamard|**20.49**|**3.02**|
> ## Response to Weakness 3:
> The theoretical target is always $Q^\star=\mathrm{msign}(W)\in\mathcal{O}(n)$; Newton-Schulz is only the numerical solver ($Q_K\approx Q^\star$). Since $W$ is initialized with Hadamard matrices (singular values near 1), the iteration operates in a favorable convergence regime. Moreover, $H$ and $H^T$ are placed in sandwiched form around linear layers (which is a trick), preventing small errors from accumulating through RMSNorm. Empirically, we verified that inference performance on GLUE is identical before and after weight fusion, confirming negligible impact. We will release the fusion code.
> ## Response to Weakness 4 & Question 3:
> We agree the current experiments are limited to GLUE. However, the mechanism remains relevant: SpikeLLM reports activation outliers two orders of magnitude larger at 7B–70B scale, suggesting saturation may worsen in larger models. Structurally, SmoothSpike is transferable—it acts on pre-activation distributions independent of task objective, and Theorem 4.1's bound $\mathcal{O}(\||x\||_2\sqrt{\log n/n})$ becomes more favorable as $n$ grows. At inference, transforms can be fused into linear layers without modifying autoregressive decoding. We will narrow claims to "mechanism-level relevance, lacking direct large-model evidence" and plan future SpikeLLM-scale experiments.
> ## Response to Weakness 5 & Question 4:
> The main contribution is improving representational quality under SNN constraints, not a strict Pareto improvement. Energy $E\propto\sum_l r_l\cdot C_l$ increases because orthogonal smoothing moves previously saturated neurons into an active-but-discriminative regime, raising overall firing rates. This is precisely why accuracy improves: more neurons operate within a useful dynamic range rather than collapsing to the maximum spike count $T$.
> ## Response to Question 1:
> Beyond smoother statistics, Figure 4 directly shows reduced $p_{\text{sat}}$ and $\Delta_{\text{sat}}$, indicating less many-to-one collapse. Table 2 shows learnable $H_1$ outperforms fixed $H_1$ by 2.14 points, confirming that data-adaptive smoothing—not generic rotation—drives the gains. This constitutes mechanism-consistent empirical support for reduced information compression.
> ## Response to Question 2:
> We measure orthogonality via $\delta_{\text{orth}}(H)=\||H^TH-I\||_F/\||I\||_F$. Results across all layers:
> ||bert|layer.0–5|layer.6–11|
> |-|-|-|-|
> |H1|0.27|–|–|
> |H2|–|0.33–0.43|0.28–0.31|
> |H3|–|0.28–0.31|0.26–0.28|
>
> All values are low. Sensitivity is minimal because the forward pass uses Newton-Schulz-projected $Q_K$, continuously pulled toward $\mathcal{O}(n)$, and the sandwiched placement prevents error accumulation.
> ## Response to Question 5:
> A fixed Hadamard applies uniform mixing and may merely transport outliers between channels, explaining its occasional underperformance. The learnable version captures two structures: (1) layer-specific outlier subspaces—redistributing energy away from saturation-sensitive directions; (2) task-relevant fidelity directions—preserving weak but discriminative features while suppressing only saturation-inducing ones. This is evidenced by fixed $H_1$ scoring 63.74 (below baseline 64.00) while learnable $H_1$ reaches 65.88. Figures 3–4 confirm the learned transform selectively attenuates long tails rather than uniformly shrinking norms.
>
> We will revise our paper based on your feedback, and we thank you again.

---

> > ### Author Rebuttal · Reviewer_DwvF · 2026-04-03
> >
> > My concerns have been adequately addressed. I appreciate the authors’ detailed clarifications and their effort to appropriately narrow the scope of the claims.

---

> > > ### Author Response · Authors · 2026-04-07
> > >
> > > Dear Reviewer DwvF,
> > >
> > > Thank you for your thorough and constructive feedback. We are pleased that our revisions have satisfactorily addressed your concerns. We deeply appreciate your positive assessment of our work and your valuable time dedicated to reviewing our manuscript. We will carefully incorporate your suggestions, along with those from other reviewers, to further strengthen the quality of the manuscript.
> > >
> > > Best regards,\
> > > The Authors

---

### Official Review · Reviewer_zbJs · 2026-03-12

**Soundness:** 4
**Presentation:** 3
**Significance:** 3
**Originality:** 4
**Overall Recommendation:** 5
**Confidence:** 4

**Summary:**

This work identifies spike saturation-induced information homogenization as a critical bottleneck in Spiking Neural Networks (SNNs), where high-amplitude inputs converge to identical spike counts under temporal constraints, truncating dynamic range.
The authors propose SmoothSpike, which smooths neuronal inputs via Hadamard transformations comprising: (1) a randomized Hadamard transform that theoretically bounds magnitudes by $\mathcal{O}(\sqrt{\frac{\log n}{n}})$, reducing saturation; and (2) a learnable orthogonal transformation (Hadamard-initialized, Newton-Schulz maintained) enabling distribution adaptation and seamless inference-time weight fusion. GLUE experiments demonstrate significant SNN-ANN gap reduction, particularly on fine-grained tasks (+16.8 CoLA, +24.8 STS-B).

**Compliance With Llm Reviewing Policy:**

Affirmed.

**Key Questions For Authors:**

1. The manuscript positions the work against multi-level spike methods. Could SmoothSpike be combined with ternary or integer-valued spikes synergistically, or do the mechanisms interfere? Have you conducted preliminary experiments in this direction?
2. Theorem 4.1 provides a probabilistic bound for the maximum element after transformation. How does this theoretical bound compare with the empirical maximum values observed in trained models across different layers (e.g., in Fig. 3)? Is the bound conservative in practice?
3. The method strictly enforces orthogonal constraints via the matrix sign function. Have you experimented with non-orthogonal learnable transformations (e.g., standard linear layers with regularization)? Is the orthogonality constraint essential for performance, or merely a convenient property for RMSNorm fusion?

**Limitations:**

yes

**Strengths And Weaknesses:**

## Strengths

* By exploiting the orthogonal invariance of RMSNorm, the method fuses transformation matrices into layer weights during inference, achieving representational gains with negligible computational overhead—an essential feature for energy-efficient neuromorphic deployment.
* The ablation studies (Table 2) systematically isolate the contributions of each transformation component ($H_1$, $H_2$, $H_3$) and demonstrate the superiority of learnable over fixed transformations.

## Weaknesses

* The evaluation is restricted to language modeling tasks (GLUE). Given that SNNs are often touted for their advantages in spatiotemporal processing, the absence of experiments on vision or audio benchmarks limits the generalizability claims.
* While Table 1 includes various baselines, the manuscript lacks direct comparison with recent multi-level or ternary spike approaches (e.g., integer LIF, ternary spikes) that also aim to enhance representational capacity, making it difficult to assess relative advantages.
* The current formulation assumes Pre-Norm Transformer architectures with RMSNorm. Its applicability to Post-Norm architectures or networks using LayerNorm/BatchNorm (where orthogonal invariance does not hold) is not demonstrated and may require architectural modifications.

---

> ### Author Rebuttal · Authors · 2026-03-29
>
> We are deeply grateful for your time in reviewing our manuscript. Your insights will be of great help to us in improving the quality of our paper.
>
> ## Response to Weakness 1:
>
> We agree that the current version focuses on GLUE and will narrow the generalizability claim in the revised manuscript. During rebuttal, we added a preliminary vision validation: we examined neuron saturation in a vision Spikingformer and found that although the average saturation rate is lower than in language models, some layers still reach ~6%, confirming that dynamic-range compression exists in vision Transformers. We fine-tuned a pretrained Spikingformer-8-768 with SmoothSpike for 50 epochs:
>
> |Method|Acc.|
> |-|-|
> |Spikingformer-8-768|75.85|
> |+SmoothSpike|**76.98**|
>
> The +1.13 gain shows SmoothSpike's benefits extend beyond language tasks. We will state this as preliminary evidence and leave broader multimodal benchmarks for future work.
>
> ## Response to Weakness 2 & Question 1:
>
> We added experiments combining SmoothSpike with SpikeLM, which already enhances representational capacity through bidirectional/ternary-style spikes. This directly tests whether SmoothSpike remains effective after multi-level spikes have been introduced.
>
> |Model|T|Energy|MNLI$_{-m/mm}$|QQP|QNLI|SST-2|CoLA|STS-B|MRPC|RTE|Avg.|
> |-|-|-|-|-|-|-|-|-|-|-|-|
> |SpikeLM|1|3.98|76.0/76.9|84.0|84.9|86.5|37.9|84.3|85.6|65.3|75.7|
> |SpikeLM|4|13.74|77.1/77.2|83.9|85.3|87.0|38.8|84.9|85.7|69.0|76.5|
> |+SmoothSpike|1|7.03|76.8/77.7|84.3|86.8|89.5|52.7|83.5|88.1|58.1|**77.5**|
>
> SmoothSpike at T=1 raises SpikeLM's average from 75.7 to 77.5 (+1.8), surpassing SpikeLM at T=4 (76.5) while consuming only 7.03 vs. 13.74 mJ. The two mechanisms are complementary: multi-level spikes expand the discrete output set per time step, while SmoothSpike improves the pre-activation distribution to mitigate saturation-induced homogenization. We believe integer-valued spikes are also compatible in principle, but leave a fully validated joint analysis for future work.
>
> ## Response to Weakness 3:
>
> We agree the paper should state the scope more clearly and will revise accordingly. Three points: (1) Most modern Transformer backbones adopt Pre-Norm, so our assumption aligns with the dominant architecture. (2) For Post-Norm or standard LayerNorm, orthogonal invariance does not hold directly due to centering and the add-then-norm path; full reuse of our zero-overhead fusion may require modifications. We will make this boundary explicit. (3) Many vision Spiking Transformers lack the add-then-norm structure, and BatchNorm can be fused into adjacent weights at inference, so SmoothSpike remains applicable—as confirmed by our Spikingformer vision results.
>
> ## Response to Question 2:
>
> We compared the theoretical bound in Theorem 4.1 with empirical maximum values:
>
> |Pre-activations|Actual Max|Theory Bound|
> |-|-|-|
> |layer.0.attn.r1|2.13|2.88|
> |layer.0.attn.r2|6.66|12.74|
> |layer.0.mlp.r1|2.07|2.96|
> |layer.0.mlp.r4|10.73|14.69|
>
> The actual/bound ratios average ~0.67, so the bound is conservative but informative: it correctly captures that Hadamard smoothing substantially suppresses extreme coordinates. This conservativeness is expected since the theorem provides a worst-case bound without exploiting data distribution or learned adaptivity. We will add this comparison and clarify its role as a mechanism-level guarantee.
>
> ## Response to Question 3:
>
> We ran a direct ablation: removing the orthogonal constraint while keeping the same Hadamard initialization and allowing unrestricted learning.
>
> |Method|Perplexity ($\downarrow$)|Eval Loss ($\downarrow$)|
> |-|-|-|
> |w/o orthogonal constraint|21.80|3.08|
> |w/ orthogonal constraint|**20.49**|**3.02**|
>
> |Matrix|$\delta_{\text{orth}}$|
> |-|-|
> |w/o orthogonal constraint|2.99|
> |w/ orthogonal constraint|0.29|
>
> Removing orthogonality degrades perplexity from 20.49 to 21.80 and increases $\delta_{\text{orth}}$ from 0.29 to 2.99. The key reason is that orthogonal transforms satisfy $\lVert Qx\rVert_2 = \lVert x\rVert_2$, performing energy redistribution rather than global scaling or distortion. This preserves the original representation geometry while dispersing outlier energy—directly aligned with SmoothSpike's core goal. A non-orthogonal transform may suppress some outliers but simultaneously distorts total energy and geometry, risking loss of task-relevant directions. Therefore, orthogonality is essential for both performance and fusion, not merely a convenient engineering choice.
>
> We will revise our paper based on your feedback, and we thank you again.

---

> > ### Author Rebuttal · Reviewer_zbJs · 2026-03-31
> >
> > My concerns have been adequately addressed.

---

> > > ### Author Response · Authors · 2026-04-07
> > >
> > > Dear Reviewer zbJs,
> > >
> > > Thank you for your thorough and constructive feedback. We are pleased that our revisions have satisfactorily addressed your concerns. We deeply appreciate your positive assessment of our work and your valuable time dedicated to reviewing our manuscript. We will carefully incorporate your suggestions, along with those from other reviewers, to further strengthen the quality of the manuscript.
> > >
> > > Best regards,\
> > > The Authors

---

### Official Review · Reviewer_jTqy · 2026-03-12

**Soundness:** 3
**Presentation:** 3
**Significance:** 3
**Originality:** 3
**Overall Recommendation:** 5
**Confidence:** 4

**Summary:**

This paper tackles "spike saturation-induced information homogenization" in Spiking Neural Networks (SNNs), where constrained time windows cause distinct high-amplitude inputs to generate identical spike outputs. The proposed SmoothSpike method employs randomized Hadamard transformations to smooth neuronal inputs, theoretically bounding extreme values to $\mathcal{O}(\sqrt{\frac{\log n}{n}})$. Extended to learnable orthogonal transformations (Hadamard-initialized with orthogonality preserved for weight fusion), SmoothSpike achieves substantial gains on the GLUE benchmark (up to +8.2 average points for Spikingformer), markedly improving fine-grained semantic discrimination.

**Compliance With Llm Reviewing Policy:**

Affirmed.

**Final Justification:**

The rebuttal addressed my main concerns.

**Key Questions For Authors:**

1. How does the energy-performance trade-off of SmoothSpike compare against alternative strategies for increasing representational capacity, such as increasing the time window T  or employing ternary spikes?

2. Figure 1 demonstrates saturation across layers, but how does the severity of information homogenization vary with different time window sizes $T$? Is the method more beneficial for shorter or longer time windows?

3. Have you evaluated SmoothSpike on non-language tasks (e.g., image classification on CIFAR-10/100 or ImageNet)? If saturation is a fundamental issue in SNNs, similar gains should theoretically manifest in vision transformers. If not, what architectural modifications would be required?

**Limitations:**

yes

**Strengths And Weaknesses:**

Strengths:

1. The paper provides a rigorous characterization of spike saturation-induced information homogenization, identifying the truncation of dynamic range as a critical factor limiting SNN representational capacity. This offers a fresh perspective beyond conventional multi-level spike extensions.

2. The theoretical analysis is compelling, with Theorem 4.1 establishing probabilistic bounds on maximum element magnitudes post-transformation and Theorem 4.3 proving the optimality of the matrix sign function for orthogonal approximation. The connection between RMSNorm's orthogonal invariance and weight fusion is technically sound.

Weaknesses:

1. While the method improves accuracy, the energy consumption increases significantly (Spikingformer: 6.76 mJ → 9.45 mJ). The paper does not thoroughly analyze whether this trade-off is favorable compared to simply increasing time steps or network depth.

2. The impact of the time window size $T$ on saturation severity and method efficacy is not systematically investigated. Additionally, the rationale for the specific choice of Newton-Schulz coefficients ($a=3.4445$, $b=-4.7750$, $c=2.0315$) lacks empirical sensitivity analysis.

3. Although related works on ternary spikes and integer-valued neurons are cited, the experimental comparison lacks direct benchmarking against these methods under identical conditions, making it difficult to assess whether SmoothSpike offers superior efficiency or accuracy relative to simply expanding spike levels.

---

> ### Author Rebuttal · Authors · 2026-03-29
>
> We are deeply grateful for your time in reviewing our manuscript. Your insights will be of great help to us in improving the quality of our paper.
>
> ## Response to Weakness 1:
>
> We added controlled experiments comparing SmoothSpike against increasing $T$ and increasing depth. For SpikeLM, enlarging $T$ from 1 to 4 yields only +0.8 avg. performance with 3.45× energy (13.74 mJ). SmoothSpike at $T=1$ achieves avg. 77.5 (+1.0 higher than SpikeLM-$T$=4) with only 7.03 mJ (48.8% lower). For Spikingformer, deepening from 4 to 12 layers gives avg. 66.8 at 6.76 mJ; SmoothSpike on 4 layers achieves 67.9 at only 2.70 mJ (+1.1 higher, 60.1% lower energy). Under SNN constraints, SmoothSpike provides a superior performance–energy trade-off compared with simply extending $T$ or depth.
> |Model|T|L|Energy|MNLI $_{-m/mm}$ |QQP|QNLI|SST-2|CoLA|STS-B|MRPC|RTE|Avg.|
> |-|-|-|-|-|-|-|-|-|-|-|-|-|
> |SpikeLM|1|12|3.98|76.0/76.9|84.0|84.9|86.5|37.9|84.3|85.6|65.3|75.7|
> |SpikeLM|4|12|13.74|77.1/77.2|83.9|85.3|87.0|38.8|84.9|85.7|69.0|76.5|
> |+SmoothSpike|1|12|7.03|76.8/77.7|84.3|86.8|89.5|52.7|83.5|88.1|58.1|**77.5**|
>
> |Model|T|L|Energy|MNLI $_{-m/mm}$ |QQP|QNLI|SST-2|CoLA|STS-B|MRPC|RTE|Avg.|
> |-|-|-|-|-|-|-|-|-|-|-|-|-|
> |Spikingformer|4|4|1.54|67.1/67.4|76.8|70.7|83.7|19.1|51.8|82.3|57.0|64.0|
> |Spikingformer|4|12|6.76|71.9/72.5|84.7|76.0|87.2|24.4|54.5|79.7|55.6|66.8|
> |+SmoothSpike|4|4|2.70|71.2/71.4|80.9|78.6|85.6|19.9|65.3|82.2|56.0|**67.9**|
>
> ## Response to Weakness 2 & Question 2:
>
> **(A) Effect of $T$ on saturation.** As $T$ increases, saturation ratio drops (7.6%→3.4%→2.4%), but the variance of inputs to still-saturated neurons increases (0.53→0.68→0.93). This means homogenization shifts from widespread to concentrated in harder outlier cases. SmoothSpike benefits all $T$: perplexity drops by 26.5%/22.9%/36.0% at $T$=1/2/4 respectively. Increasing $T$ and SmoothSpike are complementary: the former enlarges spike budget, the latter reshapes pre-activation distributions.
>
> |T|Sat. Ratio%|Var|
> |-|-|-|
> |1|7.6|0.53|
> |2|3.4|0.68|
> |4|2.4|0.93|
>
> |Method|T|PPL($\downarrow$)|eval loss($\downarrow$)|
> |-|-|-|-|
> |Spikingformer|1|44.91|3.83|
> |+SmoothSpike|1|**33.01**|**3.51**|
> |Spikingformer|2|33.27|3.49|
> |+SmoothSpike|2|**25.65**|**3.24**|
> |Spikingformer|4|31.75|3.46|
> |+SmoothSpike|4|**20.49**|**3.02**|
>
> **(B) Newton-Schulz coefficient sensitivity.** We conducted OAT perturbation and joint LHS sampling ($\pm15$ %, 160 samples). Under $\pm1$ % perturbation, all three coefficients show 0% divergence. Beyond $\pm2.5$ % $\sim5$ %, error rises sharply with ~50% divergence. In the $\pm15$ % neighborhood, 45.6% of samples diverge. The chosen coefficients lie in a locally stable, high-quality region—not arbitrary, though not necessarily globally optimal. We will describe them as a practical operating point in the revision.
>
> |Metric|Result|
> |-|-|
> |`final_orth_error@T5` of the baseline coefficients (3.4445,-4.7750,2.0315)|0.4451|
> |Total neighborhood samples|160|
> |Divergent samples / ratio|73/45.62%|
> |Non-divergent samples|87|
>
> |Coefficient|`perf_change` range under $\pm1$% perturbation|divergence rate under $\pm1$% perturbation|Representative unstable directions (leaving the local neighborhood)|
> |-|-|-|-|
> |a|[-8.63%,+8.53%]|0%|$+2.5$ % $\rightarrow$ $+394.21$% (50% divergence), $+5$ % $\rightarrow$ $+390.96$ % (50% divergence)|
> |b|[-10.74%,+11.62%]|0%|$-2.5$ % $\rightarrow$ $+386.74$ % (49.2% divergence), $-5$ % $\rightarrow$ $+370.14$ % (50% divergence)|
> |c|[-4.69%,+4.42%]|0%|$+5$ % $\rightarrow$ $+387.08$ % (49.2% divergence)|
>
> ## Response to Weakness 3 & Question 1:
>
> SpikeLM already employs ternary/bidirectional spikes, so our SpikeLM experiments directly test whether SmoothSpike benefits models that have already expanded spike levels. Mechanistically, ternary spikes expand per-step output values, while SmoothSpike reshapes pre-activation distributions via orthogonal transform—the two are complementary. Even with ternary spikes, the representable range remains bounded under finite $T$, so saturation-induced homogenization persists. Adding SmoothSpike to SpikeLM at $T$=1 achieves avg. 77.5 at 7.03 mJ, outperforming SpikeLM-$T$=4 (avg. 76.5 at 13.74 mJ) by +1.0 with 48.8% lower energy. For integer-valued neurons, SmoothSpike is theoretically compatible but training stability in language models remains under investigation; we leave this to future work.
>
> ## Response to Question 3:
>
> We examined saturation in a vision Spikingformer: the average saturation rate is lower than in language models but still reaches ~6% in some layers. Fine-tuning the pretrained Spikingformer-8-768 with SmoothSpike for 50 epochs improves Top-1 accuracy from 75.85 to 76.98 (+1.13). This preliminary result confirms SmoothSpike generalizes beyond language tasks. Full training from scratch is a planned next step.
>
> |Method|Acc.|
> |-|-|
> |Spikingformer-8-768|75.85|
> |+SmoothSpike|**76.98**|
>
> We will revise our paper based on your feedback, and we thank you again.

---

> > ### Author Rebuttal · Reviewer_jTqy · 2026-04-04
> >
> > Thank you for the detailed and thoughtful rebuttal， which has addressed my concerns. Therefore, I will maintain my positive score (Accept).

---

> > > ### Author Response · Authors · 2026-04-07
> > >
> > > Dear Reviewer jTqy,
> > >
> > > Thank you for your thorough and constructive feedback. We are pleased that our revisions have satisfactorily addressed your concerns. We deeply appreciate your positive assessment of our work and your valuable time dedicated to reviewing our manuscript. We will carefully incorporate your suggestions, along with those from other reviewers, to further strengthen the quality of the manuscript.
> > >
> > > Best regards,\
> > > The Authors

---

### Decision · Program_Chairs · 2026-04-30

**Decision:**

Accept (spotlight)

**Comment:**

**Summary of Contributions:**
This paper identifies and addresses a critical bottleneck in Spiking Neural Networks (SNNs) termed "spike saturation-induced information homogenization," where high-amplitude inputs collapse into identical maximum spike counts under restricted time windows. To mitigate this, the authors propose SmoothSpike, which utilizes a learnable Hadamard transformation to smooth extreme neuronal inputs. This prevents the truncation of the dynamic range and significantly improves the representational capacity of the model.

**Reviewer Consensus:**
The reviewing committee was unanimously positive, with all reviewers ultimately recommending acceptance. The committee agreed that the paper offers a fresh, well-grounded perspective on a concrete failure mode in SNNs and provides a conceptually clean solution.

**Key Strengths Highlighted by Reviewers:**
* **Theoretical Rigor:** Reviewers praised the theoretical bounds establishing how randomized Hadamard transformations effectively constrain extreme values and reduce saturation probability.
* **Zero-Overhead Inference:** The method exploits the orthogonal invariance of RMSNorm, allowing the transformation matrices to be fused directly into layer weights during inference. This ensures performance gains without added computational overhead.
* **Empirical Gains:** SmoothSpike demonstrated substantial performance improvements on the GLUE benchmark, particularly on fine-grained semantic tasks (e.g., CoLA and STS-B).

**Rebuttal Impact:**
The initial reviews shared a common concern regarding the narrow scope of the evaluation (limited strictly to GLUE language tasks) and the lack of direct comparison with existing multi-level/ternary spike enhancements.

The authors provided a highly effective rebuttal that successfully addressed these concerns by:
1.  Providing preliminary experimental results on vision tasks (image classification) that confirmed the saturation issue exists beyond language models and that SmoothSpike effectively improves accuracy in those domains as well.
2.  Demonstrating through new experiments that SmoothSpike is fully complementary to existing multi-level spike models (like SpikeLM), offering a superior performance-energy trade-off compared to simply extending spike levels or model depth.

**Conclusion:**
This is a technically sound, highly original paper that introduces a mathematically backed and practically efficient solution to a well-identified problem in SNNs. The authors' thorough and constructive rebuttal resolved the minor lingering doubts, making this a strong contribution to the field.